# Covariance Matrix Adaptation MAP-Annealing

## Abstract

Single-objective optimization algorithms search for the single highest-quality solution with respect to an objective. Quality diversity (QD) algorithms, such as Covariance Matrix Adaptation MAP-Elites (CMA-ME), search for a collection of solutions that are both high-quality with respect to an objective and diverse with respect to specified measure functions. However, CMA-ME suffers from three major limitations highlighted by the QD community: prematurely abandoning the objective in favor of exploration, struggling to explore flat objectives, and having poor performance for low-resolution archives. We propose a new quality diversity algorithm, Covariance Matrix Adaptation MAP-Annealing (CMA-MAE), that addresses all three limitations. We provide theoretical justifications for the new algorithm with respect to each limitation. Our theory informs our experiments, which support the theory and show that CMA-MAE achieves state-of-the-art performance.

## 1 Introduction

Consider an example problem of searching for celebrity faces in the latent space of a generative model. As a single-objective optimization problem, we specify an objective $f$ that targets a celebrity such as Tom Cruise. A single-objective optimizer, such as CMA-ES (Hansen, 2016), will converge to a single solution of high objective value, an image that looks like Tom Cruise as much as possible.

However, this objective has ambiguity. How old was Tom Cruise in the photo? Did we want the person in the image to have short or long hair? By instead framing the problem as a quality diversity optimization problem, we additionally specify a measure function $m_1$ that quantifies age and a measure function $m_2$ that quantifies hair length. A quality diversity algorithm (Pugh et al., 2015; Chatzilygeroudis et al., 2021), such as CMA-ME (Fontaine et al., 2020), can then optimize for a collection of images that are diverse with respect to age and hair length, but all look like Tom Cruise.

While previous work (Fontaine et al., 2020; 2021a;b; Earle et al., 2021) has shown that CMA-ME solves such QD problems efficiently, three important limitations of the algorithm have been discovered. First, on difficult to optimize objectives, variants of CMA-ME will abandon the objective too soon (Tjanaka et al., 2022), and instead favor exploring the measure space, the vector space defined by the measure function outputs. Second, the CMA-ME algorithm struggles to explore flat objective functions (Paolo et al., 2021). Third, CMA-ME works well on high-resolution archives, but struggles to explore low-resolution archives (Cully, 2021; Fontaine & Nikolaidis, 2021a). We note that the chosen archive resolution affects the performance of all current QD algorithms.

We propose a new algorithm, CMA-MAE, that addresses these three limitations.

To address the first limitation, we derive an algorithm that smoothly blends between CMA-ES and CMA-ME. First, consider how CMA-ES and CMA-ME differ. At each step CMA-ES's objective ranking maximizes the objective function $f$ by approximating the natural gradient of $f$ at the current solution point (Akimoto et al., 2010). In contrast, CMA-ME's improvement ranking moves in the direction of the natural gradient of $f - f_A$ at the current solution point, where $f_A$ is a discount function equal to the objective of the best solution so far that has the same measure values as the current solution point. The function $f - f_A$ quantifies the gap between a candidate solution and the best solution so far at the candidate solution's position in measure space.

Our key insight is to *anneal the function $f_A$ by a learning rate $\alpha$*. We observe that when $\alpha = 0$, then our discount function $f_A$ never increases and our algorithm behaves like CMA-ES. However, when

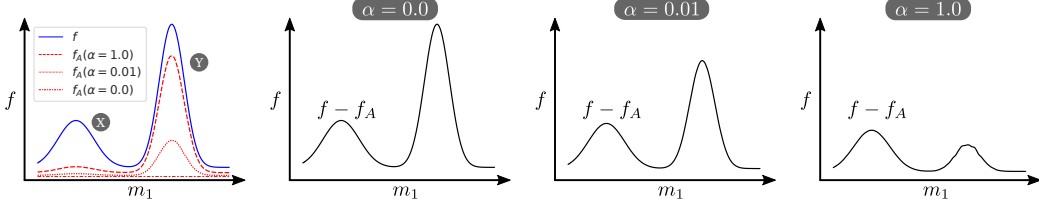

Figure 1: An example of how different $\alpha$ values affect the function $f - f_A$ optimized by CMA-MAE after a fixed number of iterations. Here $f$ is a bimodal objective where mode $X$ is harder to optimize than mode $Y$, requiring more optimization steps, and modes $X$ and $Y$ are separated by measure $m_1$. For $\alpha = 0$, the objective $f$ is equivalent to $f - f_A$, as $f_A$ remains constant. For larger values of $\alpha$, CMA-MAE discounts region $Y$ in favor of prioritizing the optimization of region $X$.

$\alpha = 1$, then our discount function always maintains the best solution for each region in measure space and our algorithm behaves like CMA-ME. For $0 < \alpha < 1$, CMA-MAE smoothly blends between the two algorithms' behavior, allowing for an algorithm that spends more time on the optimization of $f$ before transitioning to exploration. Figure 1 is an illustrative example of varying the learning rate $\alpha$.

Our proposed annealing method naturally addresses the flat objective limitation. Observe that both CMA-ES and CMA-ME struggle on flat objectives $f$ as the natural gradient becomes $\mathbf{0}$ in this case and each algorithm will restart. However, we show that, when CMA-MAE optimizes $f - f_A$ for $0 < \alpha < 1$, the algorithm becomes a descent method on the density histogram defined by the archive.

Finally, CMA-ME's poor performance on low resolution archives is likely caused by the non-stationary objective $f - f_A$ changing too quickly for the adaptation mechanism to keep up. Our archive learning rate $\alpha$ controls how *quickly* $f - f_A$ changes. We derive a conversion formula for $\alpha$ that allows us to derive equivalent $\alpha$ for different archive resolutions. Our conversion formula guarantees that CMA-MAE is the first QD algorithm invariant to archive resolution.

Overall, our work shows how a simple algorithmic change to CMA-ME addresses all three major limitations affecting CMA-ME's performance and robustness. Our theoretical findings justify the aforementioned properties and inform our experiments, which show that CMA-MAE outperforms state-of-the-art QD algorithms and maintains robust performance across different archive resolutions.

## 2 PROBLEM DEFINITION

**Quality Diversity.** We adopt the quality diversity (QD) problem definition from Fontaine & Nikolaidis (2021a). A QD problem consists of an objective $f : \mathbb{R}^n \to \mathbb{R}$ that maps $n$-dimensional solution parameters to a scalar value denoting the quality of the solution and $k$ measures $m_i : \mathbb{R}^n \to \mathbb{R}$ or, as a vector function, $\mathbf{m} : \mathbb{R}^n \to \mathbb{R}^k$ that quantify behavior or attributes of each solution[1]. The range of $\mathbf{m}$ forms a measure space $S = \mathbf{m}(\mathbb{R}^n)$. The QD objective is to find a set of solutions $\boldsymbol{\theta} \in \mathbb{R}^n$, such that $\mathbf{m}(\boldsymbol{\theta}) = \mathbf{s}$ for each $\mathbf{s}$ in $S$ and $f(\boldsymbol{\theta})$ is maximized.

The measure space $S$ is continuous, but solving algorithms need to produce a finite collection of solutions. Therefore, QD algorithms in the MAP-Elites (Mouret & Clune, 2015; Cully et al., 2015) family relax the QD objective by discretizing the space $S$. Given $T$ as the tessellation of $S$ into $M$ cells, the QD objective becomes to find a solution $\boldsymbol{\theta_i}$ for each of the $i \in \{1, \ldots, M\}$ cells, such that each $\boldsymbol{\theta_i}$ maps to the cell corresponding to $\mathbf{m}(\boldsymbol{\theta_i})$ in the tesselation $T$. The QD objective then becomes maximizing the objective value $f(\boldsymbol{\theta_i})$ of all cells:

$$\max \sum_{i=1}^{M} f(\boldsymbol{\theta_i}) \tag{1}$$

The *differentiable quality diversity* (DQD) problem (Fontaine & Nikolaidis, 2021a) is a special case of the QD problem where both the objective $f$ and measures $m_i$ are *first-order differentiable*.

---

[1]In agent-based settings, such as reinforcement learning, the measure functions are sometimes called behavior functions and the outputs of each measure function are called behavioral characteristics or behavior descriptors.

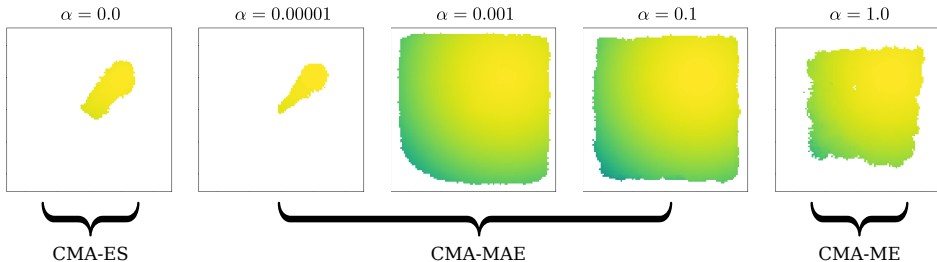

Figure 2: Our proposed CMA-MAE algorithm smoothly blends between the behavior of CMA-ES and CMA-ME via an archive learning rate $\alpha$. Each heatmap visualizes an archive of solutions across a 2D measure space, where the color of each cell represents the objective value of the solution.

## 3 PRELIMINARIES

We present several QD algorithms that solve derivative-free QD problems to provide context for our proposed CMA-MAE algorithm. Appendix D contains information about the DQD algorithm CMA-MEGA, which solves problems where exact gradient information is available.

**MAP-Elites and MAP-Elites (line)**. The MAP-Elites QD algorithm produces an archive of solutions, where each cell in the archive corresponds to the provided tesselation $T$ in the QD problem definition. The algorithm initializes the archive by sampling solutions from the solution space $\mathbb{R}^n$ from a fixed distribution. After initialization, MAP-Elites produces new solutions by selecting occupied cells uniformly at random and perturbing them with isotropic Gaussian noise: $\theta' = \theta_i + \sigma\mathcal{N}(\mathbf{0}, I)$. For each new candidate solution $\theta'$, the algorithm computes an objective $f(\theta')$ and measures $m(\theta')$. MAP-Elites places $\theta'$ into the archive if the cell corresponding to $m(\theta')$ is empty or $\theta'$ obtains a better objective value $f(\theta')$ than the current occupant. The MAP-Elites algorithm results in an archive of solutions that are diverse with respect to the measure function $m$, but also high quality with respect to the objective $f$. Vassiliades & Mouret (2018) proposed the MAP-Elites (line) algorithm by augmenting the isotropic Gaussian perturbation with a linear interpolation between two solutions $\theta_i$ and $\theta_j$: $\theta' = \theta_i + \sigma_1\mathcal{N}(\mathbf{0}, I) + \sigma_2\mathcal{N}(\mathbf{0}, 1)(\theta_i - \theta_j)$.

**CMA-ME**. Covariance Matrix Adaptation MAP-Elites (CMA-ME) (Fontaine et al., 2020) combines the archiving mechanisms of MAP-Elites with the adaptation mechanisms of CMA-ES Hansen (2016). Instead of perturbing archive solutions with Gaussian noise, CMA-ME maintains a multivariate Gaussian of search directions $\mathcal{N}(\mathbf{0}, \Sigma)$ and a search point $\theta \in \mathbb{R}^n$. The algorithm updates the archive by sampling $\lambda$ solutions around the current search point $\theta_i \sim \mathcal{N}(\theta, \Sigma)$. After updating the archive, CMA-ME ranks solutions via a two stage ranking. Solutions that discover a new cell are ranked by the objective $\Delta_i = f(\theta_i)$, and solutions that map to an occupied cell $e$ are ranked by the improvement over the incumbent solution $\theta_e$ in that cell: $\Delta_i = f(\theta_i) - f(\theta_e)$. CMA-ME prioritizes exploration by ranking all solutions that discover a new cell before all solutions that improve upon an existing cell. Finally, CMA-ME moves $\theta$ towards the largest improvement in the archive, according to the CMA-ES update rules. Fontaine & Nikolaidis (2021a) showed that the improvement ranking of CMA-ME approximates a natural gradient of a modified QD objective (see Eq. 1).

## 4 PROPOSED ALGORITHMS

We present the CMA-MAE algorithm. While we focus on CMA-MAE, the same augmentations apply to CMA-MEGA to form the novel CMA-MAEGA algorithm (see Appendix D).

**CMA-MAE.** CMA-MAE is an algorithm that adjusts the rate the objective $f - f_A$ changes. First, consider at a high level how CMA-ME explores the measure space and discovers high quality solutions. The CMA-ME algorithm maintains a solution point $\theta$ and an archive $A$ with previously discovered solutions. When CMA-ME samples a new solution $\theta'$, the algorithm computes the solution's objective value $f(\theta')$ and maps the solution to a cell $e$ in the archive based on the measure $m(\theta')$. CMA-ME then computes the improvement of the objective value $f(\theta')$ of the new solution, over a discount function $f_A : \mathbb{R}^n \to \mathbb{R}$. In CMA-ME, we define $f_A(\theta')$ by computing the cell $e$ in

the archive corresponding to $m(\boldsymbol{\theta'})$ and letting $f_A(\boldsymbol{\theta'}) = f(\boldsymbol{\theta_e})$, where $\boldsymbol{\theta_e}$ is the incumbent solution of cell $e$. The algorithm ranks candidate solutions by improvement $f(\boldsymbol{\theta'}) - f_A(\boldsymbol{\theta'}) = f(\boldsymbol{\theta'}) - f(\boldsymbol{\theta_e})$ and moves the search in the direction of higher ranked solutions.

Assume that CMA-ME samples a new solution $\boldsymbol{\theta'}$ with a high objective value of $f(\boldsymbol{\theta'}) = 99$. If the current occupant $\boldsymbol{\theta_e}$ of the corresponding cell has a low objective value of $f(\boldsymbol{\theta_e}) = 0.3$, then the improvement in the archive $\Delta = f(\boldsymbol{\theta'}) - f(\boldsymbol{\theta_e}) = 98.7$ is high and the algorithm will move the search point $\boldsymbol{\theta}$ towards $\boldsymbol{\theta'}$. Now, assume that in the next iteration the algorithm discovers a new solution $\boldsymbol{\theta''}$ with objective value $f(\boldsymbol{\theta''}) = 100$ that maps to the same cell as $\boldsymbol{\theta'}$. The improvement then is $\Delta = f(\boldsymbol{\theta''}) - f(\boldsymbol{\theta'}) = 1$ as $\boldsymbol{\theta'}$ replaced $\boldsymbol{\theta_e}$ in the archive in the previous iteration. CMA-ME would likely move $\boldsymbol{\theta}$ away from $\boldsymbol{\theta''}$ as the solution resulted in low improvement. In contrast, CMA-ES would move towards $\boldsymbol{\theta''}$ as it ranks only by the objective $f$, ignoring previously discovered solutions with similar measure values.

In the above example, CMA-ME moves away from high performing solutions in order to maximize how the archive changes. However, in domains with hard-to-optimize objective functions, it is beneficial to perform more optimization steps in high-performing regions (Tjanaka et al., 2022).

Like CMA-ME, CMA-MAE maintains a discount function $f_A(\boldsymbol{\theta'})$ and ranks solutions by improvement $f(\boldsymbol{\theta'}) - f_A(\boldsymbol{\theta'})$. However, instead of setting $f_A(\boldsymbol{\theta'})$ equal to $f(\boldsymbol{\theta_e})$, we set $f_A(\boldsymbol{\theta'}) = t_e$, where $t_e$ is an acceptance threshold maintained for each cell in the archive $A$. When adding a candidate solution to the archive, we control the rate that $t_e$ changes by the archive learning rate $\alpha$ as follows: $t_e \leftarrow (1 - \alpha)t_e + \alpha f(\boldsymbol{\theta'})$.

The archive learning rate $\alpha$ in CMA-MAE allows us to control how quickly we leave a high-performing region of measure space. For example, consider discovering solutions in the same cell with objective value 100 in 5 consecutive iterations. The improvement values computed by CMA-ME would be $100, 0, 0, 0, 0$, thus CMA-ME would move rapidly away from this cell. The improvement values computed by CMA-MAE with $\alpha = 0.5$ would diminish smoothly as follows: $100, 50, 25, 12.5, 6.25$, enabling further exploitation of the high-performing region.

Next, we walk through the CMA-MAE algorithm step-by-step. Algorithm 1 shows the pseudo-code for CMA-MAE with the differences from CMA-ME highlighted in yellow. First, on line 2 we initialize the acceptance threshold to $min_f$. In each iteration we sample $\lambda$ solutions around the current search point $\boldsymbol{\theta}$ (line 5). For each candidate solution $\boldsymbol{\theta_i}$, we evaluate the solution and compute the objective value $f(\boldsymbol{\theta_i})$ and measure values $m(\boldsymbol{\theta_i})$ (line 6). Next, we compute the cell $e$ in the archive that corresponds to the measure values and the improvement $\Delta_i$ over the current threshold $t_e$ (lines 7-8). If the objective crosses the acceptance threshold $t_e$, we replace the incumbent $\boldsymbol{\theta_e}$ in the archive and increase the acceptance threshold $t_e$ (lines 9-11). Next, we rank all candidate solutions $\boldsymbol{\theta_i}$ by their improvement $\Delta_i$. Finally, we step our search point $\boldsymbol{\theta}$ and adapt our covariance matrix $\Sigma$ towards the direction of largest improvement (lines 14-15) according to CMA-ES's update rules (Hansen, 2016).

**CMA-MAEGA.** We note that our augmentations to the CMA-ME algorithm only affects how we replace solutions in the archive and how we calculate $\Delta_i$. CMA-ME and CMA-MEGA replace solutions and calculate $\Delta_i$ identically, so we apply the same augmentations to CMA-MEGA to form a new DQD algorithm, CMA-MAEGA, in Appendix D.

## 5 THEORETICAL PROPERTIES OF CMA-MAE

We provide insights about the behavior of CMA-MAE for different $\alpha$ values. We include all proofs in Appendix E. CMA-MAEGA has similar theoretical properties discussed in Appendix F.

**Theorem 5.1.** *The CMA-ES algorithm is equivalent to CMA-MAE when $\alpha = 0$, if CMA-ES restarts from an archive solution.*

The next theorem states that CMA-ME is equivalent to CMA-MAE when $\alpha = 1$ with the following caveats: First, we assume that CMA-ME restarts only by the CMA-ES restart rules, rather than the additional "no improvement" restart rule in prior work (Fontaine et al., 2020). Second, we assume that both CMA-ME and CMA-MAE leverage $\mu$ selection (Hansen, 2016) rather than filtering selection (Fontaine et al., 2020).

**Algorithm 1** Covariance Matrix Adaptation MAP-Annealing (CMA-MAE)

---

**CMA-MAE** ($evaluate, \boldsymbol{\theta_0}, N, \lambda, \sigma, \boxed{min_f, \alpha}$)

    **input :** An evaluation function $evaluate$ that computes the objective and measures, an initial solution $\boldsymbol{\theta_0}$, a desired number of iterations $N$, a branching population size $\lambda$, an initial step size $\sigma$, a minimal acceptable solution quality $min_f$, and an archive learning rate $\alpha$.

    **result :** Generate $N\lambda$ solutions storing elites in an archive $A$.

1    Initialize solution parameters $\boldsymbol{\theta}$ to $\boldsymbol{\theta_0}$, CMA-ES parameters $\Sigma = \sigma I$ and $\boldsymbol{p}$, where we let $\boldsymbol{p}$ be the CMA-ES internal parameters.

2    Initialize the archive A and the acceptance threshold $t_e$ with $min_f$ for each cell $e$.

3    **for** $iter \leftarrow 1$ **to** $N$ **do**

4      **for** $i \leftarrow 1$ **to** $\lambda$ **do**

5        $\boldsymbol{\theta_i} \sim \mathcal{N}(\boldsymbol{\theta}, \Sigma)$

6        $f, \boldsymbol{m} \leftarrow \text{evaluate}(\boldsymbol{\theta_i})$

7        $e \leftarrow \text{calculate\_cell}(A, \boldsymbol{m})$

8        $\Delta_i \leftarrow f - t_e$

9        **if** $f > t_e$ **then**

10          Replace the current occupant in cell $e$ of the archive $A$ with $\boldsymbol{\theta_i}$

11          $t_e \leftarrow (1 - \alpha)t_e + \alpha f$

12        **end**

13      **end**

14      rank $\boldsymbol{\theta_i}$ by $\Delta_i$

15      Adapt CMA-ES parameters $\boldsymbol{\theta}, \Sigma, \boldsymbol{p}$ based on improvement ranking $\Delta_i$

16      **if** *CMA-ES converges* **then**

17        Restart CMA-ES with $\Sigma = \sigma I$.

18        Set $\boldsymbol{\theta}$ to a randomly selected existing cell $\boldsymbol{\theta_i}$ from the archive

19      **end**

20    **end**

---

**Theorem 5.2.** *The CMA-ME algorithm is equivalent to CMA-MAE when $\alpha = 1$ and $min_f$ is an arbitrarily large negative number.*

We next provide theoretical insights on how the discount function $f_A$ smoothly increases from a constant function $min_f$ to the discount function used by CMA-ME, as $\alpha$ increases from 0 to 1. We focus on the special case of a fixed sequence of candidate solutions.

**Theorem 5.3.** *Let $\alpha_i$ and $\alpha_j$ be two archive learning rates for archives $A_i$ and $A_j$ such that $0 \leq \alpha_i < \alpha_j \leq 1$. For two runs of CMA-MAE that generate the same sequence of $m$ candidate solutions $\{S\} = \boldsymbol{\theta_1}, \boldsymbol{\theta_2}, ..., \boldsymbol{\theta_m}$, it follows that $f_{A_i}(\boldsymbol{\theta}) \leq f_{A_j}(\boldsymbol{\theta})$ for all $\boldsymbol{\theta} \in \mathbb{R}^n$.*

Finally, we wish to provide insights about the exploration properties of CMA-MAE for an archive learning rate $\alpha$ between 0 and 1, when the objective $f$ is constant. Consider an approximate density descent algorithm that is identical to CMA-ME, but differs by how solutions are ranked. Specifically, we assume that this algorithm maintains a density histogram of the occupancy counts $o_e$ for each cell $e$, with $o_e$ representing the number of times a solution was generated in that cell. This algorithm descends the density histogram by ranking solutions based on the occupancy count of the cell that the solution maps to, where solutions that discover less frequently visited cells are ranked higher.

**Theorem 5.4.** *The CMA-MAE algorithm optimizing a constant objective function $f(\boldsymbol{\theta}) = C$ for all $\boldsymbol{\theta} \in \mathbb{R}^n$ is equivalent to the approximate density descent algorithm, when $0 < \alpha < 1$ and $min_f < C$.*

While Theorem 5.4 assumes a constant objective $f$, we conjecture that the theorem holds true generally when threshold $t_e$ in each cell $e$ approaches the local optimum within the cell boundaries.

## 6 EXPERIMENTS

We compare the performance of CMA-MAE with the state-of-the-art QD algorithms MAP-Elites, MAP-Elites (line), and CMA-ME, using existing Pyribs (Tjanaka et al., 2021) QD library implementations. We set $\alpha = 0.01$ for CMA-MAE and include additional experiments for varying $\alpha$

| Algorithm | LP (sphere) | | LP (Rastrigin) | | LP (plateau) | | Arm Repertoire | | LSI | |
|---|---|---|---|---|---|---|---|---|---|---|
| | QD-score | Coverage | QD-score | Coverage | QD-score | Coverage | QD-score | Coverage | QD-score | Coverage |
| MAP-Elites | 41.64 | 50.80% | 31.43 | 47.88% | 47.07 | 47.07% | 71.40 | 74.09% | 12.85 | 19.42% |
| MAP-Elites (line) | 49.07 | 60.42% | 38.29 | 56.51% | 52.20 | 52.20% | 74.55 | 75.61% | 14.40 | 21.11% |
| CMA-ME | 36.50 | 42.82% | 38.02 | 53.09% | 34.54 | 34.54% | 75.82 | 75.89% | 14.00 | 19.57% |
| CMA-MAE | **64.86** | **83.31%** | **52.65** | **80.46%** | **79.27** | **79.29%** | **79.03** | **79.24%** | **17.67** | **25.08%** |

Table 1: Mean QD-score and coverage values after 10,000 iterations for each algorithm per domain.

in section 7. Because annealing methods replace solutions based on the threshold, we retain the best solution in each cell for comparison purposes. We include additional comparisons between CMA-MEGA and CMA-MAEGA – the gradient-based counterpart of CMA-MAE – in Appendix K.

We select the benchmark domains from Fontaine & Nikolaidis (2021a): linear projection (Fontaine et al., 2020), arm repertoire (Cully & Demiris, 2017), and latent space illumination (Fontaine et al., 2021b). To evaluate the good exploration properties of CMA-MAE on flat objectives, we introduce a variant of the linear projection domain to include a "plateau" objective function that is constant everywhere for solutions within a fixed range and has a quadratic penalty for solutions outside the range. We describe the domains in detail in Appendix B.

## 6.1 EXPERIMENT DESIGN

**Independent Variables.** We follow a between-groups design with two independent variables: the algorithm and the domain.

**Dependent Variables.** We use the sum of $f$ values of all cells in the archive, defined as the QD-score Pugh et al. (2015), as a metric for the quality and diversity of solutions. Following Fontaine & Nikolaidis (2021a), we normalize the QD-score metric by the archive size (the total number of cells from the tesselation of measure space) to make the metric invariant to archive resolution. We additionally compute the coverage, defined as the number of occupied cells in the archive divided by the total number of cells.

## 6.2 ANALYSIS

Table 1 shows the QD-score and coverage values for each algorithm and domain, averaged over 20 trials for the linear projection (LP) and arm repertoire domains and over 5 trials for the LSI domain. Fig. 3 shows the QD-score values for increasing number of iterations and example archives for CMA-MAE and CMA-ME, with 95% confidence intervals.

We conducted a two-way ANOVA to examine the effect of the algorithm and domain (LP (sphere), LP (Rastrigin), LP (plateau), arm repertoire, and LSI) on the QD-score. There was a significant interaction between the search algorithm and the domain ($F(12, 320) = 1958.34, p < 0.001$). Simple main effects analysis with Bonferroni corrections showed that CMA-MAE outperformed all baselines in all benchmark domains.

For the arm repertoire domain, we can compute the optimal archive coverage by testing whether each cell overlaps with a circle of radius equal to the maximum arm length (see Appendix B). We observe that CMA-MAE approaches the computed optimal coverage $80.24\%$ for a resolution of $100 \times 100$ and outperforms CMA-MEGA (Fontaine & Nikolaidis, 2021a) (see Appendix K).

These results show that the archive learning rate $\alpha$ is particularly beneficial for CMA-MAE. We observe that CMA-MAE initially explores regions of the measure space that have high-objective values. Once the archive becomes saturated, CMA-MAE reduces to approximate density descent, as we prove in Theorem 5.4 for flat objectives. On the other hand, CMA-ME does not receive any exploration signal when the objective landscape becomes flat, resulting in poor performance.

While our results show improved quantitative results on the LSI domain, Appendix I discusses how to improve the visual quality by leveraging techniques from the generative art community. Fig. 4 shows an example collage generated by adopting improvements for guiding StyleGAN with CLIP.

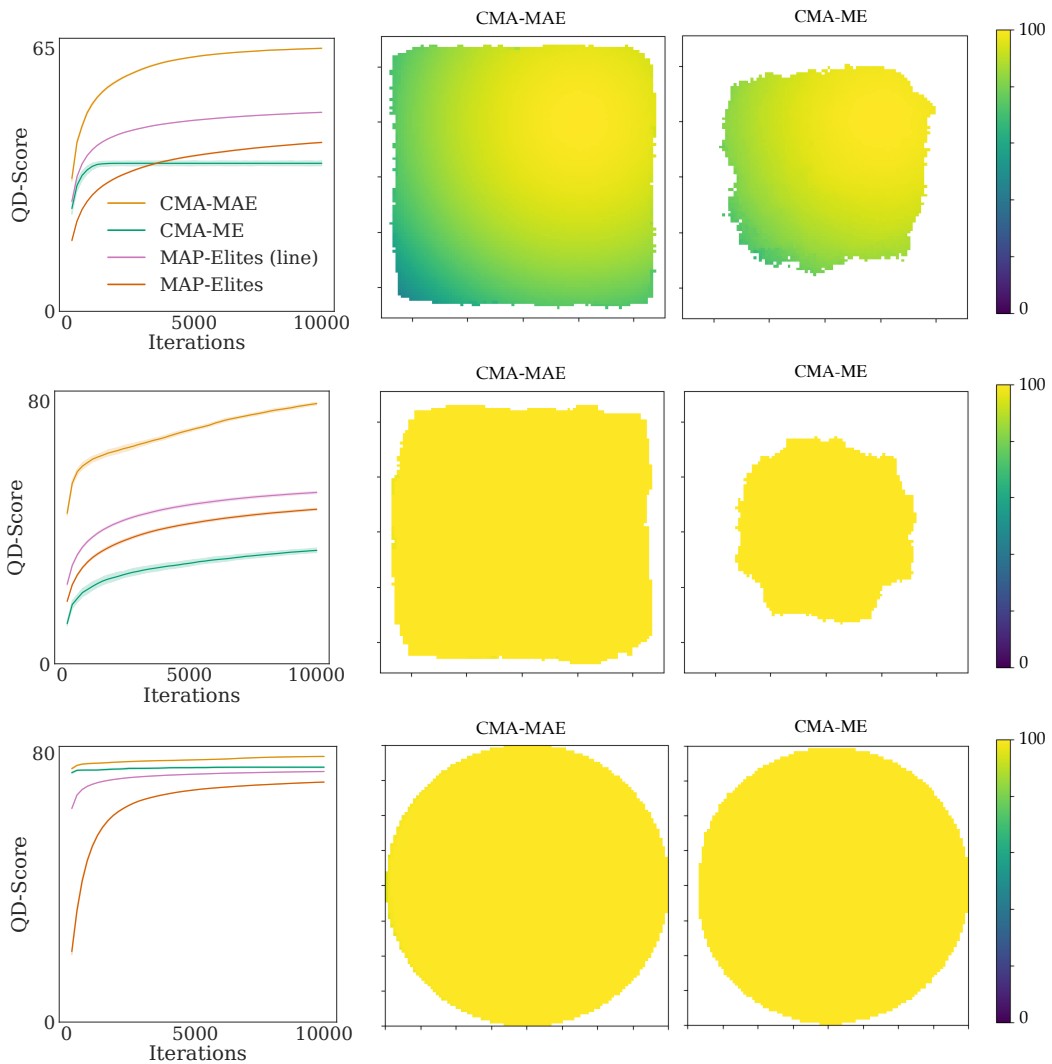

Figure 3: QD-score plot with 95% confidence intervals and heatmaps of generated archives by CMA-MAE and CMA-ME for the linear projection sphere (top), plateau (middle), and arm repertoire (bottom) domains. Each heatmap visualizes an archive of solutions across a 2D measure space.

| $\alpha$ (CMA-MAE) | LP (sphere) | | LP (Rastrigin) | | LP (plateau) | | Arm Repertoire | |
|---|---|---|---|---|---|---|---|---|
| | QD-score | Coverage | QD-score | Coverage | QD-score | Coverage | QD-score | Coverage |
| 0.000 | 5.82 | 6.06% | 5.33 | 6.24% | 19.49 | 19.49% | 65.91 | 66.25% |
| 0.001 | 62.65 | 79.36% | 47.87 | 68.10% | 77.60 | 77.68% | 78.63 | 79.07% |
| 0.010 | **64.86** | **83.31%** | **52.65** | **80.56%** | 79.27 | 79.29% | **79.03** | **79.24%** |
| 0.100 | 60.42 | 76.19% | 48.74 | 72.50% | **83.21** | **83.21%** | 78.74 | 78.85% |
| 1.000 | 37.01 | 43.50% | 37.86 | 52.82% | 34.00 | 34.00% | 75.94 | 76.01% |

Table 2: Mean QD metrics after 10,000 iterations for CMA-MAE at different learning rates.

# 7 ON THE ROBUSTNESS OF CMA-MAE

Next, we present two studies that evaluate the robustness of CMA-MAE across two hyperparameters that may affect algorithm performance: the archive learning rate $\alpha$ and the archive resolution.

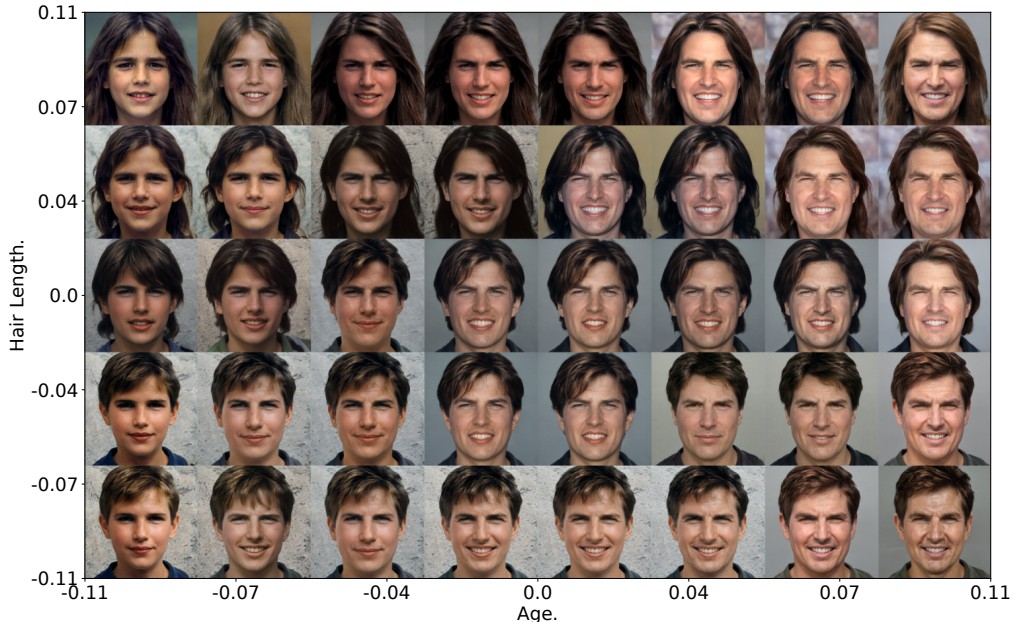

Figure 4: A latent space illumination collage for the objective "A photo of the face of Tom Cruise." with hair length and age measures. See Appendix I for more detail.

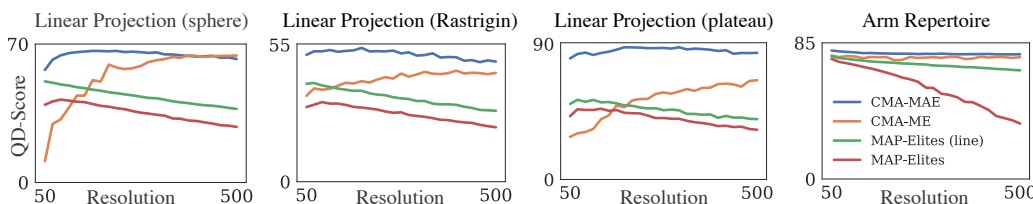

Figure 5: Final QD-score of each algorithm for 25 different archive resolutions.

**Archive Learning Rate.** We examine the effect of different archive learning rates on the performance of CMA-MAE in the linear projection and arm repertoire domains. We vary the learning rate from 0 to 1 on an exponential scale, while keeping the resolution constant in each domain.

Table 2 shows that running CMA-MAE with the different $0 < \alpha < 1$ results in relatively similar performance, showing that CMA-MAE is fairly robust to $\alpha$ values. On the other hand, if $\alpha = 0$ or $\alpha = 1$ the performance drops drastically. Setting $\alpha = 1$ results in very similar performance with CMA-ME, which supports our insight from Theorem 5.2.

**Archive Resolution.** As noted by Cully (2021) and Fontaine & Nikolaidis (2021a), quality diversity algorithms in the MAP-Elites family sometimes perform differently when run with different archive resolutions. For example, in the linear projection domain presented in Fontaine et al. (2020), CMA-ME outperformed MAP-Elites and MAP-Elites (line) for archives of resolution $500 \times 500$, while in this paper we observe that it performs worse for resolution $100 \times 100$. In this study, we investigate how CMA-MAE performs at different archive resolutions.

First, we note that the optimal archive learning rate $\alpha$ is dependent on the resolution of the archive. Consider as an example a sequence of solution additions to two archives $A_1$ and $A_2$ of resolution $100 \times 100$ and $200 \times 200$, respectively. $A_2$ subdivides each cell in $A_1$ into four cells, thus archive $A_2$'s thresholds $t_e$ should increase at a four times faster rate than $A_1$. To account for this difference, we compute $\alpha_2$ for $A_2$ via a conversion formula $\alpha_2 = 1 - (1 - \alpha_1)^r$ (see derivation in Appendix G), where $r$ is the ratio of cell counts between archives $A_1$ and $A_2$. We initialize $\alpha_1 = 0.01$ for $A_1$. In the above example, $\alpha_2 = 1 - (1 - 0.01)^4 = 0.0394$.

Fig. 5 shows the QD-score of CMA-MAE with the resolution-dependent archive learning rate and the baselines for each benchmark domain. CMA-ME performs worse as the resolution decreases because the archive changes quickly at small resolutions, affecting CMA-ME's adaptation mechanism. On the contrary, MAP-Elites and MAP-Elites (line) perform worse as the resolution increases due to having more elites to perturb. CMA-MAE's performance is invariant to the resolution of the archive.

## 8 RELATED WORK

**Quality Diversity Optimization.** The predecessor to quality diversity optimization, simply called diversity optimization, originated with the Novelty Search algorithm (Lehman & Stanley, 2011a), which searches for a collection of solutions that are diverse in measure space. Later work introduced the Novelty Search with Local Competition (NSLC) (Lehman & Stanley, 2011b) and MAP-Elites (Cully et al., 2015; Mouret & Clune, 2015) algorithms, which combined single-objective optimization with diversity optimization and were the first QD algorithms. Since then, several QD algorithms have been proposed, based on a variety of single-objective optimization methods, such as Bayesian optimization (Kent & Branke, 2020), evolution strategies (Conti et al., 2018; Colas et al., 2020; Fontaine et al., 2020), differential evolution (Choi & Togelius, 2021), and gradient ascent (Fontaine & Nikolaidis, 2021a). Several works have improved selection mechanisms (Sfikas et al., 2021; Cully & Demiris, 2017), archives (Fontaine et al., 2019; Vassiliades et al., 2018; Smith et al., 2016), and perturbation operators (Vassiliades & Mouret, 2018; Nordmoen et al., 2018).

**QD with Gradient Information.** Several works combine gradient information with quality diversity optimization in ways that do not leverage the objective and measure gradients directly. For example, in model-based quality diversity optimization (Gaier et al., 2018; Hagg et al., 2020; Cazenille et al., 2019; Keller et al., 2020; Lim et al., 2021; Zhang et al., 2021; Gaier et al., 2020), Rakicevic et al. (2021) trains an autoencoder on the archive of solutions and leverages the Jacobian of the decoder network to compute the covariance of the Gaussian perturbation. In quality diversity reinforcement learning (QD-RL), several works (Parker-Holder et al., 2020; Pierrot et al., 2020; Nilsson & Cully, 2021; Tjanaka et al., 2022) approximate a reward gradient or diversity gradient via a critic network, action space noise, or evolution strategies and incorporate those gradients into a QD-RL algorithm.

**Acceptance Thresholds.** Our proposed archive learning rate $\alpha$ was loosely inspired by simulated annealing methods (Bertsimas & Tsitsiklis, 1993) that maintain an acceptance threshold that gradually becomes more selective as the algorithm progresses. The notion of an acceptance threshold is also closely related to minimal criterion methods in evolutionary computation (Lehman & Stanley, 2010; Brant & Stanley, 2017; 2020; Stanley et al., 2016). Our work differs by both 1) maintaining an acceptance threshold per archive cell rather than a global threshold and 2) annealing the threshold.

## 9 LIMITATIONS AND FUTURE WORK

Our approach introduced two hyperparameters, $\alpha$ and $min_f$, to control the rate that $f - f_A$ changes. We observed that an $\alpha$ set strictly between $0$ and $1$ yields theoretical exploration improvements and that CMA-MAE is robust with respect to the exact choice of $\alpha$. We additionally derived a conversion formula that converts an $\alpha_1$ for a specific archive resolution to an equivalent $\alpha_2$ for a different resolution. However, the conversion formula still requires practitioners to specify a good initial value of $\alpha_1$. Future work will explore ways to automatically initialize $\alpha$, similar to how CMA-ES automatically assigns internal parameters (Hansen, 2016).

Quality diversity optimization is a rapidly growing branch of stochastic optimization with applications in generative design (Hagg et al., 2021; Gaier et al., 2020; 2018), automatic scenario generation in robotics (Fontaine & Nikolaidis, 2021c; Fontaine et al., 2021a; Fontaine & Nikolaidis, 2021b), reinforcement learning (Parker-Holder et al., 2020; Pierrot et al., 2020; Nilsson & Cully, 2021; Tjanaka et al., 2022), damage recovery in robotics (Cully et al., 2015), and procedural content generation (Gravina et al., 2019; Fontaine et al., 2021b; Zhang et al., 2021; Earle et al., 2021; Khalifa et al., 2018; Steckel & Schrum, 2021; Schrum et al., 2020; Sarkar & Cooper, 2021; Bhatt et al., 2022). Our paper introduces a new quality diversity algorithm, CMA-MAE. Our theoretical findings inform our experiments, which show that CMA-MAE addresses three major limitations affecting the CMA-ME algorithm, leading to state-of-the-art performance.

## 10 Ethics Statement

By controlling the trade-off between exploration and exploitation in QD algorithms, we aim towards improving their performance and robustness, thus making these algorithms easier to apply in a wide range of domains and applications. One promising application is synthetically extracting datasets from generative models to train machine learning algorithms Jahanian et al. (2021); Besnier et al. (2020). This can raise ethical considerations because generative models can reproduce and exacerbate existing biases in the datasets that they were trained on (Jain et al., 2020; Menon et al., 2020). On the other hand, quality diversity algorithms with carefully selected measure functions can target diversity with desired attributes, thus we hypothesize that they can be effective in generating balanced datasets. Furthermore, by attempting to find diverse solutions, QD algorithms are a step towards open-endedness in AI Stanley et al. (2017) and will often result in unexpected and often surprising emergent behaviors (Lehman et al., 2020). We recognize that this presents several challenges in predictability and monitoring of AI systems (Hendrycks et al., 2021), and we highlight the importance of future work on balancing the tradeoff between open-endedness and control (Ecoffet et al., 2020).

## 11 Reproducibility Statement

In the supplemental material we provide complete source code for all algorithms and experiments, as well as the Conda environments for installing project dependencies. The "README.md" document provides complete instructions both setup and execution of all experiments. In Appendix A we provide all hyperparameters. In Appendix B we provide domain-specific details for replicating all experimental domains. In Appendix C we provide information about the computational resources and hardware we used to run our experiments. In Appendix D we provide the pseudocode for the CMA-MAEGA algorithm, the DQD counterpart of CMA-MAE. In Appendix E we provide the proofs of all theorems in the paper. In Appendix F we provide the theoretical properties of CMA-MAEGA. In Appendix G we provide the derivation of the conversion formula for the archive learning rate. In Appendix H we provide a batch threshold update rule that is invariant to the order that the solutions are processes within a batch update. In Appendix I we discuss the implementation details for additional experiments that improve the quality of the generated images in the latent space illumination domain. In Appendix K we present all metrics with standard errors for each algorithm and domain.

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

APPENDIX

# A    HYPERPARAMETER SELECTION

For all domains we mirror the hyperparameter selection of Fontaine & Nikolaidis (2021a). For CMA-MAE and CMA-MAEGA, we duplicate the hyperparameter selections of CMA-ME and CMA-MEGA, respectively. Following Fontaine et al. (2020), we run all algorithms with 15 emitters on the linear projection and arm repertoire domains. In the latent space illumination domain, we run experiments with only one emitter, due to the computational expense of the domain. Emitters are independent CMA-ES instances that run in parallel with a shared archive. For each algorithm, we select a batch size $\lambda = 36$ following Fontaine & Nikolaidis (2021a). For MAP-Elites and MAP-Elites (line), we initialize the archive with 100 random solutions, sampled from the distribution $\mathcal{N}(\mathbf{0}, I)$. These initial solutions do not count in the evaluation budget for MAP-Elites and MAP-Elites (line). For algorithms in the CMA-ME family (CMA-ME, CMA-MAE, CMA-MEGA, and CMA-MAEGA), we initialize $\boldsymbol{\theta_0} = \mathbf{0}$ for every domain.

In our experiments we want to directly compare the ranking mechanisms of CMA-ME and CMA-MAE. However, CMA-ME is typically run with a "no improvement" restart rule, where the algorithm will restart if no solution changes the archive. Due to CMA-MAE's annealed acceptance threshold $t_e$, a "no improvement" restart rule would cause CMA-ME and CMA-MAE to restart at different rates, confounding the effects of restarts and rankings. Filter selection also has a similar confounding effect as solutions are selected if they change the archive. For these reasons, in the main paper we run CMA-ME with a basic restart rule (CMA-ES style restarts only (Hansen, 2016)) and $\mu$ selection (Hansen, 2016) (selecting the top half of the ranking). In Appendix Section K, we run an extra CMA-ME with filter selection and the "no improvement" restart rule, which we denote CMA-ME*. We include, as an additional baseline, a configuration of CMA-ME that mixes emitters that optimize only for the objective with emitters that optimize for improvement, a configuration first studied by Cully (2021). We refer to this configuration as CMA-ME (imp, opt).

In the latent space illumination domain, due to the computational expense of the domain, we compare directly against the results from Fontaine & Nikolaidis (2021a), where we obtained the data (MIT license) with consent from the authors. For CMA-MAE and CMA-MAEGA we include the "no improvement" restart rule to match CMA-ME and CMA-MEGA as closely as possible. For this domain, we take gradient steps with the Adam optimizer (Kingma & Ba, 2015), following the recommendation of Fontaine & Nikolaidis (2021a). However, we run CMA-MAE with $\mu$ selection, since we found that small values of the archive learning rate $\alpha$ makes filter selection worse.

In Appendix I, we describe a second LSI experiment on StyleGAN2 (Karras et al., 2020b) configured by insights from the generative art community that improve the quality of single-objective latent space optimization. For this domain, we configure CMA-MAEGA and CMA-MEGA to use a "basic" restart rule because the latent space L2 regularization keeps solutions in the StyleGAN2 training distribution. For this experiment, the latent space is large ($n = 9216$), so we exclude CMA-ME and CMA-MAE due to the size of the covariance matrix ($9216 \times 9216$) and the prohibitive cost for computing an eigendecomposition of a large covariance matrix.

**Linear Projection (sphere, Rastrigin, plateau).**

- MAP-Elites: $\sigma = 0.5$
- MAP-Elites (line): $\sigma_1 = 0.5$, $\sigma_2 = 0.2$
- CMA-ME: $\sigma = 0.5$, $\mu$ selection, basic restart rule
- CMA-ME*: $\sigma = 0.5$, filter selection, no improvement restart rule
- CMA-ME (imp, opt): $\sigma = 0.5$, $\mu$ selection, basic restart rule, 7 optimizing and 8 improvement emitters
- CMA-MAE: $\sigma = 0.5$, $\alpha = 0.01$, $min_f = 0$, $\mu$ selection, basic restart rule
- CMA-MEGA: $\sigma_g = 10.0$, $\eta = 1.0$, basic restart rule, gradient ascent optimizer
- CMA-MAEGA: $\sigma_g = 10.0$, $\eta = 1.0$, $\alpha = 0.01$, $min_f = 0$, basic restart rule, gradient ascent optimizer

**Arm Repertoire.**

- MAP-Elites: $\sigma = 0.1$
- MAP-Elites (line): $\sigma_1 = 0.1$, $\sigma_2 = 0.2$
- CMA-ME: $\sigma = 0.2$, $\mu$ selection, basic restart rule
- CMA-ME*: $\sigma = 0.2$, filter selection, no improvement restart rule
- CMA-ME (imp, opt): $\sigma = 0.2$, $\mu$ selection, basic restart rule, 7 optimizing and 8 improvement emitters
- CMA-MAE: $\sigma = 0.2$, $\alpha = 0.01$, $min_f = 0$, $\mu$ selection, basic restart rule
- CMA-MEGA: $\sigma_g = 0.05$, $\eta = 1.0$, basic restart rule, gradient ascent optimizer
- CMA-MAEGA: $\sigma_g = 0.05$, $\eta = 1.0$, $\alpha = 0.01$, $\min_f = 0$, basic restart rule, gradient ascent optimizer

**Latent Space Illumination. (StyleGAN)**

- MAP-Elites: $\sigma = 0.2$
- MAP-Elites (line): $\sigma_1 = 0.1$, $\sigma_2 = 0.2$
- CMA-ME: $\sigma = 0.02$, filter selection, no improvement restart rule
- CMA-MAE: $\sigma = 0.02$, $\alpha = 0.1$, $min_f = 55$, $\mu$ selection, no improvement restart rule, 50 iteration timeout
- CMA-MEGA: $\sigma_g = 0.002$, $\eta = 0.002$, Adam optimizer, no improvement restart rule
- CMA-MAEGA: $\sigma_g = 0.002$, $\eta = 0.002$, $\alpha = 0.1$, $min_f = 55$, Adam optimizer, no improvement restart rule, 50 iteration timeout

**Latent Space Illumination. (StyleGAN 2)**

- MAP-Elites: $\sigma = 0.1$
- MAP-Elites (line): $\sigma_1 = 0.1$, $\sigma_2 = 0.2$
- CMA-MEGA: $\sigma_g = 0.01$, $\eta = 0.05$, Adam optimizer, basic restart rule
- CMA-MAEGA: $\sigma_g = 0.01$, $\eta = 0.05$, $\alpha = 0.02$, $min_f = 0$, Adam optimizer, basic restart rule

**Adam Hyperparameters.** We use the same hyperparameters as previous work Perez (2021); Fontaine & Nikolaidis (2021a).

- $\beta_1 = 0.9$
- $\beta_2 = 0.999$

**Archives.** For the linear projection and arm repertoire domains, we initialize an archive of $100 \times 100$ cells for all algorithms. For latent space illumination we initialize an archive of $200 \times 200$ cells for all algorithms, following Fontaine & Nikolaidis (2021a).

## B  DOMAIN DETAILS

To experimentally evaluate both CMA-MAE and CMA-MAEGA, we select domains from Fontaine & Nikolaidis (2021a): linear projection (Fontaine et al., 2020), arm repertoire (Cully & Demiris, 2017), and latent space illumination (Fontaine et al., 2021b). While many quality diversity optimization domains exist, we select these because gradients of $f$ and $m$ are easy to compute analytically and allow us to evaluate DQD algorithms in addition to derivative-free QD algorithms. To evaluate the good exploration properties of CMA-MAE on flat objectives, we introduce a variant of the linear projection domain to include a "plateau" objective function.

**Linear Projection.** The linear projection domain (Fontaine et al., 2020) was introduced to benchmark distortions caused by mapping a high-dimensional search space to a low-dimensional measure space.

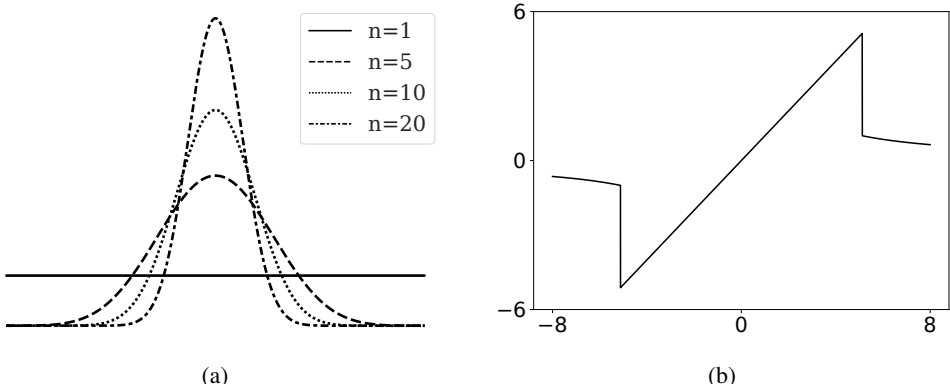

(a)            (b)

Figure 6: Measure function figures reproduced from prior work (Fontaine et al., 2020; Fontaine & Nikolaidis, 2021a) with the authors' permission. (a) a Bates distribution. (b) $clip$ function for defining the measures in the linear projection domain.

The domain forms a 2D measure space by a linear projection that bounds the contribution of each component $\theta_i$ of the projection to the range $[-5.12, 5.12]$. QD algorithms must adapt the step size of each component $\theta_i$ to slowly approach the extremes of the measure space, with a harsh penalty for components outside $[-5.12, 5.12]$. As QD domains must provide an objective, the linear projection domain included two objectives from the black-box optimization benchmarks (Hansen et al., 2016; 2010): sphere and Rastrigin. Following Fontaine et al. (2020), we run all experiments for $n = 100$.

Formally, the measure functions are defined as a linear projection, a weighted sum of the components $\theta_i \in \mathbb{R}$ of a solution $\boldsymbol{\theta} \in \mathbb{R}^n$. The first measure function $m_1$ is a weighted sum of the first half of the solution $\boldsymbol{\theta}$, and the second measure function $m_2$ is a weighted sum of the second half of the solution $\boldsymbol{\theta}$ (see Eq. 3). To ensure that all solutions mapped to measure space occupy a finite volume, the contribution in measure space of each component $\theta_i$ is bounded to the range $[-5.12, 5.12]$ via a clip function (see Eq. 2) that applies a harsh penalty for solution components $\theta_i$ stepping outside the range $[-5.12, 5.12]$.

$$clip(\theta_i) = \begin{cases} \theta_i & \text{if } -5.12 \leq \theta_i \leq 5.12 \\ 5.12/\theta_i & \text{otherwise} \end{cases} \tag{2}$$

$$m(\boldsymbol{\theta}) = \left( \sum_{i=1}^{\lfloor \frac{n}{2} \rfloor} clip(\theta_i), \sum_{i=\lfloor \frac{n}{2} \rfloor + 1}^{n} clip(\theta_i) \right) \tag{3}$$

Fig. 6 visualizes *why* the linear projection domain is challenging. First, we note that the density of solutions in search space mapped to measure space mostly occupies the region close to $\mathbf{0}$. To justify why, consider sampling uniformly in the hypercube $[-5.12, 5.12]^n$ in search space. We note that each of these points maps to the linear region of the measure functions and each of our measures becomes a sum of random variables. If we divide by $n$, we normalize by the dimensions of the search space, then the measure functions become an *average* of random variables. The average of $n$ uniform random variables is the Bates distribution (Johnson et al., 1995), a distribution that narrows in variance as $n$ grows larger. Without the $clip$ function, a QD algorithm could simply increase a single $\theta_i$ to reach any point in the measure space. However, the clip function prevents this by bounding the contribution of each component of $\boldsymbol{\theta}$ to the range $[-5.12, 5.12]$. To reach the extremes of measure space all components $\theta_i$ must converge to the extremums $\pm 5.12$. The linear projection domain is challenging to explore due to both the clustering of solutions in a small region of measure space and the heavy measure space penalties applied by the clip function when a component $\theta_i$ leaves the region $[-5.12, 5.12]$.

Next, we describe the linear projection domain's objective functions visualized in Fig. 7.

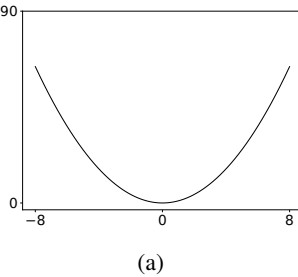 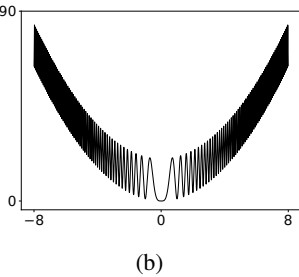 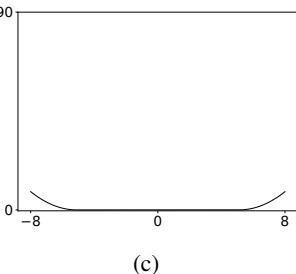

(a)                   (b)                   (c)

Figure 7: Objective functions for the linear projection domain in their minimization form for $n = 1$. (a) a sphere function. (b) the Rastrigin function. (c) a plateau function.

The objectives of the linear projection domain satisfy the requirements that a QD domain needs to have an objective and are of lesser importance than the measure function definitions, since the benchmark primarily evaluates exploration capabilities. Fontaine et al. (2020) selected two objectives from the black-box optimization benchmarks competition (Hansen et al., 2016; 2010): sphere and Rastrigin. The sphere function (Eq. 4) is a quadratic function[2], while the Rastrigin function (Eq. 5) is a multi-modal function that when smoothed is quadratic. The domain shifts the global optimum to the position $\theta_i = 5.12 \cdot 0.4 = 2.048$.

$$f_{sphere}(\boldsymbol{\theta}) = \sum_{i=1}^{n} \theta_i^2 \tag{4}$$

$$f_{Rastrigin}(\boldsymbol{\theta}) = 10n + \sum_{i=1}^{n} [\theta_i^2 - 10\cos(2\pi\theta_i^2)] \tag{5}$$

We introduce an additional objective to evaluate the good exploration properties of CMA-MAE on flat objectives. Our "plateau" objective function (Eq. 7) is constant everywhere, but with a quadratic penalty for each component outside the range $[-5.12, 5.12]$. The penalty acts as a regularizer to encourage algorithms to search in the linear region of measure space.

$$f_{plateau}(\theta_i) = \begin{cases} 0 & \text{if } -5.12 \leq \theta_i \leq 5.12 \\ (|x| - 5.12)^2 & \text{otherwise} \end{cases} \tag{6}$$

$$f_{plateau}(\boldsymbol{\theta}) = \frac{1}{n} \sum_{i=1}^{n} f_{plateau}(\theta_i) \tag{7}$$

**Arm Repertoire.** The arm repertoire domain Cully & Demiris (2017); Vassiliades & Mouret (2018) tasks QD algorithms to find a diverse collection of arm positions for an $n$-dimensional planar robotic arm with revolute joints. The measures in this domain are the 2D coordinates of the robot's end-effector and the objective is to minimize the variance of the joint angles.

In Fig. 8, we visualize example arms for $n = 5$ (5-DOF). The optimal solutions in this domain have 0 variance between all joint angles. The measure functions are bounded to the range $[-n, n]$ as each arm segment has a unit length. The reachable cells form a circle of radius $n$. Therefore, the optimal archive coverage is approximately $\frac{\pi n^2}{4n^2} \approx 78.5\%$. An archive can achieve an upper-bound of this ratio that becomes tighter at higher resolutions. We select $n = 100$ (100-DOF) arms for the experiments.

**Latent Space Illumination.** Prior work introduced the latent space illumination problem (Fontaine et al., 2021b), the problem of searching the latent space of a generative model with a quality diversity algorithm. We evaluate on the StyleGAN+CLIP version of this problem (Fontaine & Nikolaidis,

---

[2]In derivative-free optimization many of the benchmark functions are named after the shape of the contour lines. In the case of quadratic functions with an identity Hessian matrix, the contour lines form hyperspheres.

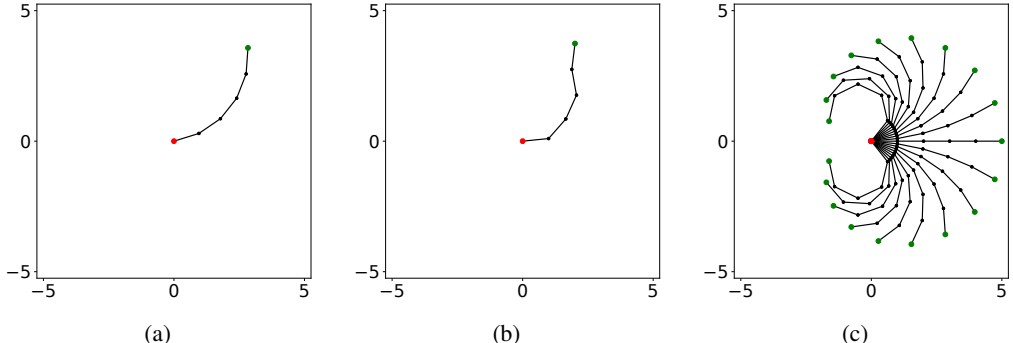

Figure 8: Examples of the Arm Repertoire domain for $n = 5$. The figures are reproduced from previous work (Fontaine & Nikolaidis, 2021a) with the authors' permission. (a) An optimal grasp with 0 variance between joint angles. (b) A sub-optimal grasp. (c) An ensemble of 0 variance optimal grasps.

2021a), by searching the latent space of StyleGAN (Karras et al., 2019) with a QD algorithm. We form the differentiable objective and measures in this domain by specifying text prompts to the CLIP model (Radford et al., 2021) that can determine the similarity of an image and text. We specify an objective prompt of "A photo of Beyonce". For measures, we would like to have CLIP quantify abstract concepts like the hair length or age of the person in the photo. However, CLIP can only determine similarity of an image and a text prompt. As surrogates for age and hair length, we specify the measure prompts of "A small child" and "A woman with long blonde hair". The objective and measure functions guide the QD algorithms towards discovering a collection of photos of Beyoncé with varying age and hair length.

For our additional LSI experiment on StyleGAN2 with setup improvements, see Appendix I.

**Transformations of the Objective Function.** We highlight two issues that must be addressed by transforming the objective in each domain. First, we note that the problem definition in each of our domains contains an objective $f$ that must be minimized. In contrast, the QD problem definition specifies an objective $f$ that must be maximized. Second, the QD-score metric, which measures the performance of QD algorithms, requires a non-negative objective function. Following prior work (Fontaine et al., 2020; Fontaine & Nikolaidis, 2021a), we transform the objective $f$ via a linear transformation: $f' = af + b$. The linear transformation maps function outputs to the range $[0, 100]$.

In the linear projection domain, we estimate the largest objective value for the sphere and Rastrigin function within the region $[-5.12, 5.12]$ for each solution component $\theta_i$. We compute $f(-5.12, -5.12, ..., -5.12)$ for each objective as the maximum. The minimum of each function is 0. We calculate the linear transformation as:

$$f'(\boldsymbol{\theta}) = 100 \cdot \frac{f(\boldsymbol{\theta}) - f_{max}}{f_{min} - f_{max}} \tag{8}$$

For our new plateau objective, all solution points within the region $[-5.12, 5.12]^n$ have objective value of 0. For this objective we set $f_{min} = 0$ and $f_{max} = 100$ and apply the transformation in Eq. 8.

For the arm domain we select $f_{min} = 0$ and $f_{max} = 1$, and in the LSI domain we select $f_{min} = 0$ and $f_{max} = 10$. We select these values to match Fontaine & Nikolaidis (2021a).

## C IMPLEMENTATION

We replicate the implementation details of prior work (Fontaine & Nikolaidis, 2021a).

**Archives.** For the linear projection and arm repertoire domains, we initialize an archive of $100 \times 100$ cells for all algorithms. For latent space illumination we initialize an archive of $200 \times 200$ cells for all algorithms, following previous work (Fontaine & Nikolaidis, 2021a).

|  | LP (sphere) | | LP (Rastrigin) | | LP (plateau) | | Arm Repertoire | |
|---|---|---|---|---|---|---|---|---|
| Algorithm | QD-score | Coverage | QD-score | Coverage | QD-score | Coverage | QD-score | Coverage |
| CMA-MEGA | 75.32 | **100.00%** | **63.07** | **100.00%** | **100.00** | **100.00%** | 75.21 | 75.25% |
| CMA-MAEGA | **75.39** | **100.00%** | 63.06 | **100.00%** | **100.00** | **100.00%** | **79.27** | **79.35%** |

Table 3: Mean QD-score and coverage values after 10,000 iterations for each DQD algorithm in the LP and arm repertoire domains.

|  | LSI (StyleGAN) | | LSI (StyleGAN2) | |
|---|---|---|---|---|
| Algorithm | QD-score | Coverage | QD-score | Coverage |
| CMA-MEGA | 16.08 | 22.58% | 9.17 | 14.91% |
| CMA-MAEGA | **16.20** | **23.83%** | **11.51** | **18.62%** |

Table 4: Mean QD-score and coverage values after 10,000 iterations for each DQD algorithm in the LSI (StyleGAN) and LSI (StyleGAN2) domains.

**Metrics.** We use the sum of $f$ values of all cells in the archive, defined as the QD-score Pugh et al. (2015), as a metric for the quality and diversity of solutions. Following Fontaine & Nikolaidis (2021a), we normalize the QD-score by the total number of cells, both occupied and unoccupied, to make QD-score invariant to the resolution of the archive. We additionally compute the coverage, defined as the number of occupied cells in the archive divided by the total number of cells.

**Computational Resources.** We ran all trials of the linear projection and arm repertoire domains on an AMD Ryzen Threadripper 32-core (64 threads) processor. A run of 20 trials in parallel takes about 20 minutes for the linear projection domain and 25 minutes for the arm repertoire domain. For the latent space illumination domain, we accelerate the StyleGAN+CLIP pipeline on a GeForce RTX 3090 Nvidia GPU. One trial for latent space illumination takes approximately 2 hours and 30 minutes for StyleGAN and approximately 3 hours and 30 minutes for StyleGAN2. In all domains, runtime increases when an algorithm obtains better coverage, because we iterate over the archive when QD statistics are calculated.

**Software Implementation.** We use the open source Pyribs (Tjanaka et al., 2021) library for all algorithms. We implemented the CMA-MAE and CMA-MAEGA algorithms using the same library.

## D   COVARIANCE MATRIX ADAPTATION MAP-ELITES VIA A GRADIENT ARBORESCENCE (CMA-MAEGA)

In this section, we provide information of the CMA-MEGA differentiable quality diversity (DQD) algorithm, and we derive CMA-MAE's DQD counterpart: CMA-MAEGA.

**CMA-MEGA.** Covariance Matrix Adaptation MAP-Elites via Gradient Arborescence (CMA-MEGA) solves the DQD problem, where the objective $f$ and measures $\boldsymbol{m}$ are first-order differentiable. Like CMA-ME, the algorithm maintains a solution point $\boldsymbol{\theta} \in \mathbb{R}^n$ and a MAP-Elites archive. CMA-MEGA samples new solutions by perturbing the search point $\boldsymbol{\theta}$ via the objective and measure gradients. However, the contribution of each gradient is balanced by gradient coefficients $\boldsymbol{c}$: $\boldsymbol{\theta_i} = \boldsymbol{\theta} + c_0 \boldsymbol{\nabla} f(\boldsymbol{\theta}) + \sum_{j=1}^{k} c_j \boldsymbol{\nabla} m_j(\boldsymbol{\theta})$. These coefficients are sampled from a multivariate Gaussian distribution $N(\boldsymbol{\mu}, \Sigma)$ maintained by the algorithm. After sampling new candidate solutions $\boldsymbol{\theta_i}$, the solutions are ranked via the improvement ranking from CMA-ME. CMA-MEGA updates $N(\boldsymbol{\mu}, \Sigma)$ via the CMA-ES update rules and the algorithm steps $\boldsymbol{\theta}$ also in the direction of largest archive improvement. The authors showed that CMA-MEGA approximates a natural gradient step of the QD objective (Eq. 1), but with respect to the gradient coefficients.

**CMA-MAEGA.** We note that our augmentations to the CMA-ME algorithm only affects how we replace solutions in the archive and how we calculate $\Delta_i$. Both CMA-ME and CMA-MAEGA replace solutions and calculate $\Delta_i$ identically, so we apply the same augmentations from CMA-ME to CMA-MEGA to form a new DQD algorithm, CMA-MAEGA. Algorithm 2 shows the pseudo-code for CMA-MAEGA with the differences from CMA-MEGA highlighted in yellow.

---

**Algorithm 2** Covariance Matrix Adaptation MAP-Annealing via a Gradient Arborescence (CMA-MAEGA)

---

CMA-MAEGA $(evaluate, \boldsymbol{\theta_0}, N, \lambda, \eta, \sigma_g, min_f, \alpha)$

    **input :** An evaluation function $evaluate$ that computes the objective, the measures, and the gradients of the objective and measures, an initial solution $\boldsymbol{\theta_0}$, a desired number of iterations $N$, a branching population size $\lambda$, a learning rate $\eta$, an initial step size for CMA-ES $\sigma_g$, a minimal acceptable solution quality $min_f$, and an archive learning rate $\alpha$.

    **result :** Generate $N(\lambda + 1)$ solutions storing elites in an archive $A$.

1    Initialize solution parameters $\boldsymbol{\theta}$ to $\boldsymbol{\theta_0}$, CMA-ES parameters $\boldsymbol{\mu} = \mathbf{0}, \Sigma = \sigma_g I$, and $\boldsymbol{p}$, where we let $\boldsymbol{p}$ be the CMA-ES internal parameters.

2    Initialize the archive $A$ and the acceptance threshold $t_e$ with $min_f$ for each cell $e$.

3    **for** $iter \leftarrow 1$ **to** $N$ **do**

4        $f, \boldsymbol{\nabla}_f, \boldsymbol{m}, \boldsymbol{\nabla_m} \leftarrow \text{evaluate}(\boldsymbol{\theta})$

5        $\boldsymbol{\nabla}_f \leftarrow \text{normalize}(\boldsymbol{\nabla}_f), \boldsymbol{\nabla}_m \leftarrow \text{normalize}(\boldsymbol{\nabla}_m)$

6        **if** $f > t_e$ **then**

7            Replace the current elite in cell $e$ of the archive $A$ with $\boldsymbol{\theta_i}$

8            $t_e \leftarrow (1 - \alpha)t_e + \alpha f$

9        **end**

10       **for** $i \leftarrow 1$ **to** $\lambda$ **do**

11          $\boldsymbol{c} \sim \mathcal{N}(\boldsymbol{\mu}, \Sigma)$

12          $\boldsymbol{\nabla}_i \leftarrow c_0 \boldsymbol{\nabla}_f + \sum_{j=1}^{k} c_j \boldsymbol{\nabla}_{m_j}$

13          $\boldsymbol{\theta'_i} \leftarrow \boldsymbol{\theta} + \boldsymbol{\nabla}_i$

14          $f', *, \boldsymbol{m'}, * \leftarrow \text{evaluate}(\boldsymbol{\theta'_i})$

15          $\Delta_i \leftarrow f' - t_e$

16          **if** $f' > t_e$ **then**

17             Replace the current occupant in cell $e$ of the archive $A$ with $\boldsymbol{\theta_i}$

18             $t_e \leftarrow (1 - \alpha)t_e + \alpha f'$

19          **end**

20       **end**

21       rank $\boldsymbol{\nabla}_i$ by $\Delta_i$

22       $\boldsymbol{\nabla}_{\text{step}} \leftarrow \sum_{i=1}^{\lambda} w_i \boldsymbol{\nabla}_{\text{rank[i]}}$

23       $\boldsymbol{\theta} \leftarrow \boldsymbol{\theta} + \eta \boldsymbol{\nabla}_{\text{step}}$

24       Adapt CMA-ES parameters $\boldsymbol{\mu}, \Sigma, \boldsymbol{p}$ based on improvement ranking $\Delta_i$

25       **if** *there is no change in the archive* **then**

26          Restart CMA-ES with $\boldsymbol{\mu} = 0, \Sigma = \sigma_g I$.

27          Set $\boldsymbol{\theta}$ to a randomly selected existing cell $\boldsymbol{\theta_i}$ from the archive

28       **end**

29    **end**

---

**Experiments.** We compare CMA-MEGA and CMA-MAEGA in the five benchmark domains. Table 3 and Table 4 shows the QD-score and coverage values for each algorithm and domain, averaged over 20 trials for the linear projection (LP) and arm repertoire domains and over 5 trials for the LSI domains. We conducted a two-way ANOVA to examine the effect of the algorithm and domain (LP (sphere), LP (Rastrigin), LP (plateau), arm repertoire, LSI (StyleGAN), and LSI (StyleGAN2) on the QD-score. There was a significant interaction between the search algorithm and the domain ($F(5, 168) = 165.7, p < 0.001$). Simple main effects analysis with Bonferroni corrections showed that CMA-MAEGA outperformed CMA-MEGA in the LP (sphere), arm repertoire, and LSI (StyleGAN2) domains. There was no statistically significance difference between the two algorithms in the LP (Rastrigin), LP (plateau), and LSI (StyleGAN) domains.

We attribute the absence of a statistical difference in the QD-score between the two algorithms on the LP (Rastrigin) and LP (plateau) domains on the perfect coverage obtained by both algorithms. Thus, any differences in QD-score are based on the objective values of the solutions returned by each algorithm. In LP (plateau), the optimal objective for each cell is easily obtainable for both methods. The LP (Rastrigin) domain contains many local optima, because of the form of the objective function

(Eq. 5). CMA-MEGA will converge to these optima before restarting, behaving as a single-objective optimizer within each local optimum. Because of the large number of local optima in the domain, it results in higher QD-score.

In the LSI (StyleGAN) domain, we attribute similar performance between CMA-MEGA and CMA-MAEGA to the restart rules used to keep each search within the training distribution of StyleGAN. On the other hand, in the LSI (StyleGAN2) domain, we regularize the search space by an L2 penalty in latent space, allowing for a larger learning rate and a basic restart rule for both algorithms, while still preventing drift out of the training distribution of StyleGAN2. Because of the fewer restarts, CMA-MAEGA can take advantage of the density descent property, which was shown to improve exploation in CMA-MAE, and outperform CMA-MEGA. We note that because StyleGAN2 has a better conditioning on the latent space (Karras et al., 2020b), it is better suited for gradient-based optimizers, which helps better distinguish between the two algorithms.

## E   THEORETICAL PROPERTIES OF CMA-MAE

**Theorem E.1.** *The CMA-ES algorithm is equivalent to CMA-MAE when $\alpha = 0$, if CMA-ES restarts from an archive solution.*

*Proof.* CMA-ES and CMA-MAE differ only on how they rank solutions. CMA-ES ranks solutions purely based on the objective $f$, while CMA-MAE ranks solutions by $f - t_e$, where $t_e$ is the acceptance threshold initialized by $min_f$. Thus, to show that CMA-ES is equivalent to CMA-MAE for $\alpha = 0$, we only need to show that they result in identical rankings.

In CMA-MAE, $t_e$ is updated as follows: $t_e \leftarrow (1 - \alpha)t_e + \alpha f$. For $\alpha = 0$, $t_e = min_f$ is invariant for the whole algorithm: $t_e \leftarrow 1t_e + 0f = t_e$. Therefore, CMA-MAE ranks solutions based on $f - min_f$. However, comparison-based sorting is invariant to order-preserving transformations of the values being sorted Hansen (2016). Thus, CMA-ES and CMA-MAE rank solutions identically.   □

Next, we prove that CMA-ME is equivalent to CMA-MAE with the following caveats. First, we assume that CMA-ME restarts only with the CMA-ES restart rules, rather than the additional "no improvement" restart condition from Fontaine et al. (2020). Second, we assume that both CMA-ME and CMA-MAE leverage $\mu$ selection rather than filtering selection.

**Lemma E.2.** *During execution of the CMA-MAE algorithm with $\alpha = 1$, the threshold $t_e$ is equal to $f(\boldsymbol{\theta_e})$ for cells that are occupied by a solution $\boldsymbol{\theta_e}$ and to $min_f$ for all empty cells.*

*Proof.* We will prove the lemma by induction. All empty cells are initialized with $t_e = min_f$, satisfying the basis step. Then, we will show that if the statement holds after $k$ archive updates, it will hold after a subsequent update $k + 1$.

Assume that at step $k$ we generate a new solution $\boldsymbol{\theta_i}$ mapped to a cell $e$. We consider two cases:

**Case 1**: The archive cell $e$ is empty. Then, $f(\boldsymbol{\theta_i}) > min_f$ and both CMA-ME and CMA-MAE will place $\boldsymbol{\theta_i}$ in the archive as the new cell occupant $\boldsymbol{\theta_e}$. The threshold $t_e$ is updated as $t_e = (1 - \alpha)t_e + \alpha f(\boldsymbol{\theta_e}) = 0min_f + 1f(\boldsymbol{\theta_e}) = f(\boldsymbol{\theta_e})$.

**Case 2**: The archive cell $e$ contains an incumbent solution $\boldsymbol{\theta_e}$. Then, either $f(\boldsymbol{\theta_i}) \leq f(\boldsymbol{\theta_e})$ or $f(\boldsymbol{\theta_i}) > f(\boldsymbol{\theta_e})$. If $f(\boldsymbol{\theta_i}) \leq f(\boldsymbol{\theta_e})$, then the archive does not change and the inductive step holds via the inductive hypothesis. If $f(\boldsymbol{\theta_i}) > f(\boldsymbol{\theta_e})$, then $\boldsymbol{\theta_i}$ becomes the new cell occupant $\boldsymbol{\theta_e}$ and $t_e$ is updated as $t_e = (1 - \alpha)t_e + \alpha f(\boldsymbol{\theta_e}) = 0t_e + 1f(\boldsymbol{\theta_e}) = f(\boldsymbol{\theta_e})$.   □

**Theorem E.3.** *The CMA-ME algorithm is equivalent to CMA-MAE when $\alpha = 1$ and $min_f$ is an arbitrarily large negative number.*

*Proof.* Both CMA-ME and CMA-MAE rank candidate solutions $\boldsymbol{\theta_i}$ based on improvement values $\Delta_i$. While CMA-ME and CMA-MAE compute $\Delta_i$ differently, we will show that for $\alpha = 1$, the rankings are identical for the two algorithms.

We assume a new candidate solution mapped to a cell $e$. We describe first the computation of $\Delta_i$ for CMA-ME. CMA-ME ranks solutions that discover an empty cell based on their objective value. Thus,

if $\boldsymbol{\theta_i}$ discovers an empty cell, $\Delta_i = f(\boldsymbol{\theta_i})$. On the other hand, if $\boldsymbol{\theta_i}$ is mapped to a cell occupied by another solution $\boldsymbol{\theta_e}$, it will rank $\boldsymbol{\theta_i}$ based on the improvement $\Delta_i = f(\boldsymbol{\theta_i}) - f(\boldsymbol{\theta_e})$. CMA-ME performs a two-stage ranking, where it ranks all solutions that discover empty cells before solutions that improve occupied cells.

We now show the computation of $\Delta_i$ for CMA-MAE with $\alpha = 1$. If $\boldsymbol{\theta_i}$ discovers an empty cell $\Delta_i = f(\boldsymbol{\theta_i}) - t_e$ and by Lemma E.2 $\Delta_i = f(\boldsymbol{\theta_i}) - min_f$. If $\boldsymbol{\theta_i}$ is mapped to a cell occupied by another solution $\boldsymbol{\theta_e}$, $\Delta_i = f(\boldsymbol{\theta_i}) - t_e$ and by Lemma E.2 $\Delta_i = f(\boldsymbol{\theta_i}) - f(\boldsymbol{\theta_e})$.

Comparing the values $\Delta_i$ between the two algorithms we observe the following: (1) If $\boldsymbol{\theta_i}$ discovers an empty cell, $\Delta_i = f(\boldsymbol{\theta_i}) - min_f$ for CMA-MAE. However, $min_f$ is a constant and comparison-based sorting is invariant to order preserving transformations (Hansen, 2016), thus ranking by $\Delta_i = f(\boldsymbol{\theta_i}) - min_f$ is identical to ranking by $\Delta_i = f(\boldsymbol{\theta_i})$ performed by CMA-ME. (2) If $\boldsymbol{\theta_i}$ is mapped to a cell occupied by another solution $\boldsymbol{\theta_e}$, $\Delta_i = f(\boldsymbol{\theta_i}) - f(\boldsymbol{\theta_e})$ for both algorithms. (3) Because $min_f$ is an arbitrarily large negative number $f(\boldsymbol{\theta_i}) - min_f > f(\boldsymbol{\theta_i}) - f(\boldsymbol{\theta_e})$. Thus, CMA-MAE will always rank solutions that discover empty cells before solutions that are mapped to occupied cells, identically to CMA-ME. $\square$

We next provide theoretical insights on how the discount function $f_A$ smoothly increases from a constant function $min_f$ to CMA-ME's discount function as $\alpha$ increases from 0 to 1. We show this for the special case of a fixed sequence of candidate solutions.

**Theorem E.4.** *Let $\alpha_i$ and $\alpha_j$ be two archive learning rates for archives $A_i$ and $A_j$ such that $0 \leq \alpha_i < \alpha_j \leq 1$. For two runs of CMA-MAE that generate the same sequence of $m$ candidate solutions $\{S\} = \boldsymbol{\theta_1}, \boldsymbol{\theta_2}, ..., \boldsymbol{\theta_m}$, it follows that $f_{A_i}(\boldsymbol{\theta}) \leq f_{A_j}(\boldsymbol{\theta})$ for all $\boldsymbol{\theta} \in \mathbb{R}^n$.*

*Proof.* We prove the theorem via induction over the sequence of solution additions. $f_A$ is the histogram formed by the thresholds $t_e$ over all archive cells $e$ in the archive. Thus, we prove $f_{A_i} \leq f_{A_j}$ by showing that $t_e(A_i) \leq t_e(A_j)$ for all archive cells $e$ after $m$ archive additions.

As a basis step, we note that $A_i$ equals $A_j$ as both archives are initialized with $min_f$.

Our inductive hypothesis states that after $k$ archive additions we have $t_e(A_i) \leq t_e(A_j)$, and we need to show that $t_e(A_i) \leq t_e(A_j)$ after solution $\boldsymbol{\theta_{k+1}}$ is added to each archive.

Our solution $\boldsymbol{\theta_{k+1}}$ has three cases with respect to the acceptance thresholds:

**Case 1:** $f(\boldsymbol{\theta_{k+1}}) \leq t_e(A_i) \leq t_e(A_j)$. The solution is not added to either archive and our property holds from the inductive hypothesis.

**Case 2:** $t_e(A_i) \leq f(\boldsymbol{\theta_{k+1}}) \leq t_e(A_j)$. The solution is added to $A_i$, but not $A_j$, thus $t'_e(A_j) = t_e(A_j)$. We follow the threshold update: $t'_e(A_i) = (1 - \alpha_i)t_e(A_i) + \alpha_i f(\boldsymbol{\theta_{k+1}})$. Next, we need to show that $t'_e(A_i) \leq t'_e(A_j)$ to complete the inductive step:

$$
\begin{aligned}
(1 - \alpha_i)t_e(A_i) + \alpha_i f(\boldsymbol{\theta_{k+1}}) &\leq f(\boldsymbol{\theta_{k+1}}) &&\iff \\
(1 - \alpha_i)t_e(A_i) &\leq (1 - \alpha_i)f(\boldsymbol{\theta_{k+1}}) &&\iff \\
t_e(A_i) &\leq f(\boldsymbol{\theta_{k+1}}) \quad \text{as } 1 - \alpha_i \geq 0
\end{aligned}
$$

The last inequality holds true per our initial assumption for Case 2. From the inductive hypothesis, we have $f(\boldsymbol{\theta_{k+1}}) \leq t_e(A_j) = t'_e(A_j)$.

**Case 3:** $t_e(A_i) \leq t_e(A_j) \leq f(\boldsymbol{\theta_{k+1}})$. The solution $\boldsymbol{\theta_{k+1}}$ is added to both archives. We need to show that $t'_e(A_i) \leq t'_e(A_j)$:

$$
\begin{aligned}
t'_e(A_i) &\leq t'_e(A_j) &&\iff \\
(1 - \alpha_i)t_e(A_i) + \alpha_i f(\boldsymbol{\theta_{k+1}}) &\leq (1 - \alpha_j)t_e(A_j) + \alpha_j f(\boldsymbol{\theta_{k+1}}) &&\text{(9)}
\end{aligned}
$$

We can rewrite Eq. 9 as:

$$
(1 - \alpha_j)t_e(A_j) - (1 - \alpha_i)t_e(A_i) + \alpha_j f(\boldsymbol{\theta_{k+1}}) - \alpha_i f(\boldsymbol{\theta_{k+1}}) \geq 0 \tag{10}
$$

First, note that:

$$(1 - \alpha_j)t_e(A_j) - (1 - \alpha_i)t_e(A_i) \geq (1 - \alpha_j)t_e(A_i) - (1 - \alpha_i)t_e(A_i)$$
$$= (1 - \alpha_j - 1 + \alpha_i)t_e(A_i)$$
$$= (\alpha_i - \alpha_j)t_e(A_i).$$

Thus:

$$(1 - \alpha_j)t_e(A_j) - (1 - \alpha_i)t_e(A_i) \geq (\alpha_i - \alpha_j)t_e(A_i) \tag{11}$$

From Eq. 10 and 11 we have:

$$(1 - \alpha_j)t_e(A_j) + \alpha_j f(\boldsymbol{\theta_{k+1}}) - (1 - \alpha_i)t_e(A_i) - \alpha_i f(\boldsymbol{\theta_{k+1}})$$
$$\geq (\alpha_i - \alpha_j)t_e(A_i) + (\alpha_j - \alpha_i)f(\boldsymbol{\theta_{k+1}})$$
$$= (\alpha_j - \alpha_i)(f(\boldsymbol{\theta_{k+1}}) - t_e(A_i))$$

As $\alpha_j > \alpha_i$ and $f(\boldsymbol{\theta_{k+1}}) \geq t_e(A_i)$, we have $(\alpha_j - \alpha_i)(f(\boldsymbol{\theta_{k+1}}) - t_e(A_i)) \geq 0$. This completes the proof that Eq. 10 holds.

As all cases in our inductive step hold, our proof by induction is complete. □

Next, we wish to provide insights about the exploration properties of CMA-MAE for an archive learning rate $\alpha$ between 0 and 1, when the objective $f$ is constant. Consider an approximate density descent algorithm that is identical to CMA-ME, but differs by how solutions are ranked. Specifically, the algorithm maintains a histogram of occupancy counts $o_e$ for each cell $e$, with $o_e$ representing the number of times a solution was generated in that cell. This algorithm descends the density histogram by ranking solutions based on the occupancy count of the cell that the solution maps to, where solutions that discover less frequently visited cells are ranked higher.

**Lemma E.5.** *The threshold $t_e$ after $k$ additions to cell $e$ forms a strictly increasing sequence for a constant objective function $f(\boldsymbol{\theta}) = C$ for all $\boldsymbol{\theta} \in \mathbb{R}^n$, when $0 < \alpha < 1$ and $min_f < C$.*

*Proof.* To show that $t_e$ after $k$ additions to cell $e$ forms a strictly increasing sequence, we write a recurrence relation for $t_e$ after $k$ solutions have been added to cell $e$. Let $t_e(k) = (1 - \alpha)t_e(k - 1) + \alpha f(\theta_i)$ and $t_e(0) = min_f$ be that recurrence relation. To show the recurrence is an increasing function, we need to show that $t_e(k) > t_e(k - 1)$ for all $k \geq 0$.

We prove the inequality via induction over cell additions $k$. As a basis step, we show $t_e(1) > t_e(0)$: $(1 - \alpha)min_f + \alpha C > min_f \iff min_f - min_f - \alpha \cdot min_f + \alpha C \iff \alpha C > \alpha \cdot min_f$. As $C > min_f$ and $\alpha > 0$, the basis step holds.

For the inductive step, we assume that $t_e(k) > t_e(k - 1)$ and need to show that $t_e(k + 1) > t_e(k)$: $t_e(k + 1) > t_e(k) \iff (1 - \alpha)t_e(k) + \alpha C > (1 - \alpha)t_e(k - 1) + \alpha C \iff (1 - \alpha)t_e(k) > (1 - \alpha)t_e(k - 1) \iff t_e(k) > t_e(k - 1)$. □

**Theorem E.6.** *The CMA-MAE algorithm optimizing a constant objective function $f(\boldsymbol{\theta}) = C$ for all $\boldsymbol{\theta} \in \mathbb{R}^n$ is equivalent to the approximate density descent algorithm, when $0 < \alpha < 1$ and $min_f < C$.*

*Proof.* We will prove that for an arbitrary archive $A$ with both the occupancy count for each cell $o_e$ and the threshold value $t_e$ computed with arbitrary learning rate $0 < \alpha < 1$, CMA-MAE results in the same ranking for an arbitrary batch of solutions $\{\boldsymbol{\theta_i}\}$ as the approximate density descent algorithm.

We let $\boldsymbol{\theta_i}$ and $\boldsymbol{\theta_j}$ be two arbitrary solutions in the batch mapped to cells $e_i$ and $e_j$. Without of loss of generality, we let $o_{e_i} \leq o_{e_j}$. The approximate density descent algorithm will thus rank $\boldsymbol{\theta_i}$ before $\boldsymbol{\theta_j}$. We will show that CMA-MAE results in the same ranking.

If $o_{e_i} \leq o_{e_j}$, and since $t_e$ is a strictly increasing function from Lemma E.5: $t_{e_i}(o_{e_i}) \leq t_{e_j}(o_{e_j})$. We have $t_{e_i}(o_{e_i}) \leq t_{e_j}(o_{e_j}) \iff C - t_{e_i}(o_{e_i}) \geq C - t_{e_j}(o_{e_j})$. Thus, the archive improvement by adding $\boldsymbol{\theta_i}$ to the archive is larger than the improvement by adding $\boldsymbol{\theta_j}$ and CMA-MAE will rank $\boldsymbol{\theta_i}$ higher than $\boldsymbol{\theta_j}$, identically with density descent. □

While Theorem E.6 assumes a constant objective $f$, we conjecture that the theorem holds true generally when threshold $t_e$ in each cell $e$ approaches the local optimum within the cell boundaries.

**Conjecture E.7.** *The CMA-MAE algorithm becomes equivalent to the density descent algorithm for a subset of archive cells for an arbitrary convex objective $f$, where the cardinality of the subset of cells increases as the number of iterations increases.*

We provide intuition for our conjecture through the lense of the elite hypervolume hypothesis (Vassiliades & Mouret, 2018). The elite hypervolume hypothesis states that optimal solutions for the MAP-Elites archive form a connected region in search space. Later work (Rakicevic et al., 2021), connected the elite hypervolume hypothesis to the manifold hypothesis (Fefferman et al., 2016) in machine learning, stating that the elite hypervolume can be represented by a low dimensional manifold in search space.

For our conjecture, we assume that the elite hypervolume hypothesis holds and there exists a smooth manifold that represents the hypervolume. Next, we assume in the conjecture that $f$ is an arbitrary convex function. As $f$ is convex, early in the CMA-MAE search the discount function $f_A$ will be flat and the search point $\boldsymbol{\theta}$ will approach the global optimum following CMA-ES's convergence properties (Hansen & Ostermeier, 1997; Hansen et al., 2003), where the precision of convergence is controlled by archive learning rate $\alpha$. By definition, the global optimum $\boldsymbol{\theta}^*$ is within the elite hypervolume as no other solution of higher quality exists within its archive cell. Assuming the elite hypervolume hypothesis holds, a subset of adjacent solutions in search space will also be in the hypervolume due to the connectedness of the hypervolume. As $f_A$ increases around the global optimum, we conjecture that the function $f(\boldsymbol{\theta}^*) - f_A(\boldsymbol{\theta}^*)$ will form a plateau around the optimum, since it will approach the value $f(\boldsymbol{\theta_i}) - f_A(\boldsymbol{\theta_i})$ of adjacent solutions $\boldsymbol{\theta_i}$. By Theorem E.6 we have a density descent algorithm within the plateau, pushing CMA-MAE to discover solutions on the frontier of the known hypervolume.

Finally, we remark that our conjecture implies that $f - f_A$ tends towards a constant function in the limit, resulting in a density descent algorithm across the elite hypervolume manifold as the number of generated solutions approaches infinity. We leave a formal proof of this conjecture for future work.

# F   Theoretical Properties of CMA-MAEGA

In this section, we investigate how the theoretical properties of CMA-MAE apply to CMA-MAEGA. While many of the properties are nearly a direct mapping, we note that, while CMA-MAE is equivalent to the single-objective optimization algorithm CMA-ES for $\alpha = 0$, there is no single-objective counterpart to CMA-MAEGA. To make the direct mapping easier, we introduce a counterpart: the gradient arborescence ascent algorithm.

The gradient arborescence ascent algorithm is similar to CMA-MEGA, but without an archive. Like CMA-MEGA, the algorithm assumes a differentiable objective $f$ and differentiable measures $\boldsymbol{m}$. However, the algorithm leverages the objective and measure function gradients only to improve the optimization of the objective $f$, rather than to find solutions that are diverse with respect to measures $\boldsymbol{m}$. As with CMA-MEGA, the gradient arborescence algorithm branches in objective-measure space. However, the algorithm ranks solutions purely by the objective function $f$ and adapts the coefficient distribution $N(\boldsymbol{\mu}, \Sigma)$ towards the natural gradient of the objective $f$.

Next, we prove properties of CMA-MAEGA that directly follow from the properties of CMA-MAE.

**Theorem F.1.** *The gradient arborescence ascent algorithm is equivalent to CMA-MAEGA when $\alpha = 0$, if gradient arborescence ascent restarts from an archive elite.*

*Proof.* We note that CMA-MAEGA and the gradient arborescence ascent algorithm differ only in how they rank solutions, and we note that the differences between CMA-MAE and CMA-ES mirror the differences between CMA-MAEGA and gradient arborescence ascent algorithm. So by directly adapting the proof of Theorem E.1, we complete our proof. □

**Theorem F.2.** *The CMA-MEGA algorithm is equivalent to CMA-MAEGA when $\alpha = 1$ and $min_f$ is an arbitrarily large negative number.*

*Proof.* We note that CMA-MAEGA and the CMA-MEGA algorithm differ only in how they rank solutions and how they update the archive $A$, and we note that the differences between CMA-MAE and CMA-ME mirror the differences between CMA-MAEGA and CMA-MEGA. So by directly adapting the proof of Theorem E.3, we complete our proof. □

**Theorem F.3.** *Let $\alpha_i$ and $\alpha_j$ be two archive learning rates for archives $A_i$ and $A_j$ such that $0 \leq \alpha_i < \alpha_j \leq 1$. For two runs of CMA-MAEGA that generate the same sequence of $m$ candidate solutions $\{S\} = \theta_1, \theta_2, ..., \theta_m$, it follows that $f_{A_i}(\theta) \leq f_{A_j}(\theta)$ for all $\theta \in \mathbb{R}^n$.*

*Proof.* We note that CMA-MAE and CMA-MAEGA update the archive $A$ in exactly the same way. Therefore, the proof follows directly by adapting the proof of Theorem E.4 to CMA-MAEGA. □

Next, we wish to show that CMA-MAEGA results in density descent in measure space. However, we need a counterpart to the approximate density descent algorithm we defined in Theorem E.6.

Consider an approximate density descending arborescence algorithm that is identical to CMA-MEGA, but differs by how solutions are ranked. Specifically, we assume that this algorithm maintains an occupancy count $o_e$ for each cell $e$, which is the number of times a solution was generated in that cell. The density descent algorithm ranks solutions based on the occupancy count of the cell that the solution maps to, where solutions that discover less frequently visited cells are ranked higher. The algorithm takes steps in search space $\mathbb{R}^n$ that minimize the approximate density function defined by the archive and adapts the coefficient distribution $N(\mu, \Sigma)$ towards coefficients that minimize the density function.

**Theorem F.4.** *The CMA-MAEGA algorithm optimizing a constant objective function $f(\theta) = C$ for all $\theta \in \mathbb{R}^n$ is equivalent to the approximate density descending arborescence algorithm, when $0 < \alpha < 1$ and $\min_f < C$.*

*Proof.* The proof of Theorem E.6 relies only on how CMA-MAE updates the archive $A$ and acceptance threshold $t_e$. The proof of this theorem follows directly by adapting the proof of Theorem E.6 to CMA-MAEGA. □

# G  DERIVATION OF THE CONVERSION FORMULA FOR THE ARCHIVE LEARNING RATE

In this section, we derive the archive learning rate conversion formula $\alpha_2 = 1 - (1 - \alpha_1)^r$ mentioned in Section 7 of the main paper, where $r$ is the ratio between archive cell counts, and $\alpha_1$ and $\alpha_2$ are archive learning rates for two archives $A_1$ and $A_2$.

Given an archive learning rate $\alpha_1$ for $A_1$, we want to derive an equivalent archive learning rate $\alpha_2$ for $A_2$ that results in robust performance when CMA-MAE is run with either $A_1$ or $A_2$. A principled way to derive a conversion formula for $\alpha_2$ is to look for an invariance property that affects the performance of CMA-MAE and that holds when CMA-MAE generates solutions in archives $A_1$ and $A_2$.

Since CMA-MAE ranks solutions by $f - f_A$, we wish for $f_A$ to increase at the same rate in the two archives. Since $f_A(\theta) = t_e$, where $t_e$ is the cell that a solution $\theta$ maps to, we select the *average value* of the acceptance thresholds $t_e$ over all cells in each archive as our invariant property.

We assume an arbitrary sequence of $N$ solution additions $\theta_1, \theta_2, ..., \theta_N$, evenly dispersed across the archive cells. We then specify $t_e$ as a function that maps $k$ cell additions to a value $t_e$ in archive cell $e$.[3] Equation 12 then defines the average value of $t_e$ across the archive after $N$ additions to an archive $A$ with $M$ cells.

$$\frac{1}{M} \sum_{i=1}^{M} t_e \left( \frac{N}{M} \right) \tag{12}$$

Then, equation 13 defines the invariance we want to guarantee between archives $A_1$ and $A_2$.

---

[3] Here we abuse notation and view $t_e$ as a function instead of threshold for simplicity and to highlight the connection to the threshold value $t_e$.

$$\frac{1}{M_1} \sum_{i=1}^{M_1} t_e\left(\frac{N}{M_1}\right) = \frac{1}{M_2} \sum_{i=1}^{M_2} t_e\left(\frac{N}{M_2}\right) \tag{13}$$

In Eq. 13, we let $M_1$ and $M_2$ the number of cells in archives $A_1$ and $A_2$, and we assume that $M_1$ and $M_2$ divide $N$. To solve for a closed form of $\alpha_2$ subject to our invariance, we need a formula for the function $t_e$. Similar to Lemma E.5, we can represent the function $t_e$ as a recurrence relation after adding $k$ solutions to cell $e$ of an archive $A$.

$$t_e(0) = min_f$$
$$t_e(k) = (1 - \alpha)t_e(k - 1) + \alpha f(\boldsymbol{\theta_k}) \tag{14}$$

Next, we look to derive a closed form for $t_e(k)$ for an archive $A$ as a way to manipulate Equation 13. However, solving for $t_e(k)$ when $f$ is an arbitrary function is difficult, because different regions of the archive will change at different rates. Instead, we solve for the special case when $f(\boldsymbol{\theta}) = C$ and $min_f < C$, where $C \in \mathbb{R}$ is a constant scalar. To solve for a closed form of the recurrence $t_e(k)$, we leverage the recurrence unrolling method (Graham et al., 1989), allowing us to guess the closed form in Equation 15.

$$
\begin{aligned}
t_e(1) &= (1 - \alpha)t_e(0) + \alpha C = (1 - \alpha)min_f + \alpha C \\
t_e(2) &= (1 - \alpha)t_e(1) + \alpha C = (1 - \alpha)[(1 - \alpha)min_f + \alpha C] + \alpha C \\
&= (1 - \alpha)^2 min_f + (1 - \alpha)\alpha C + \alpha C \\
t_e(3) &= (1 - \alpha)t_e(2) + \alpha C \\
&= (1 - \alpha)[(1 - \alpha)^2 min_f + (1 - \alpha)\alpha C + \alpha C] + \alpha C \\
&= (1 - \alpha)^3 min_f + (1 - \alpha)^2 \alpha C + (1 - \alpha)\alpha C + \alpha C \\
&\vdots \\
t_e(k) &= (1 - \alpha)^k min_f + \sum_{i=0}^{k-1}(1 - \alpha)^i \alpha C
\end{aligned}
\tag{15}
$$

We recognize the summation in Equation 15 as a geometric series. As $0 < \alpha < 1$, we rewrite the summation as follows.

$$
\begin{aligned}
t_e(k) &= (1 - \alpha)^k min_f + \sum_{i=0}^{k-1}(1 - \alpha)^i \alpha C \\
&= (1 - \alpha)^k min_f + \alpha C\left(\frac{1 - (1 - \alpha)^k}{1 - (1 - \alpha)}\right) \\
&= (1 - \alpha)^k min_f + \alpha C\left(\frac{1 - (1 - \alpha)^k}{\alpha}\right) \\
&= (1 - \alpha)^k min_f + C - C(1 - \alpha)^k \\
&= (min_f - C)(1 - \alpha)^k + C \\
&= C - (C - min_f)(1 - \alpha)^k
\end{aligned}
\tag{16}
$$

Next, we prove that the closed form we guessed is the closed form of the recurrence relation.

**Theorem G.1.** *The recurrence relation $t_e(0) = min_f$ and $t_e(k) = (1 - \alpha)t_e(k - 1) + \alpha C$ has the closed form $t_e(k) = C - (C - min_f)(1 - \alpha)^k$, where $0 < \alpha < 1$ and $min_f < C$.*

*Proof.* We show the closed form holds via induction over cell additions $k$.

As a basis step we show that $t_e(0) = C - (C - min_f)(1 - \alpha)^0 = C - (C - min_f) = min_f$.

For the inductive step, suppose after $j$ insertions into the archive $A$ in cell $e$ our closed form holds. We show that the closed form holds for $j + 1$ insertions.

$$
\begin{aligned}
t_e(j + 1) &= (1 - \alpha)t_e(j) + \alpha C \\
&= (1 - \alpha)[C - (C - min_f)(1 - \alpha)^j] + \alpha C \\
&= C(1 - \alpha) - (C - min_f)(1 - \alpha)^{j+1} + \alpha C \\
&= C - \alpha C + \alpha C - (C - min_f)(1 - \alpha)^{j+1} \\
&= C - (C - min_f)(1 - \alpha)^{j+1}
\end{aligned}
\tag{17}
$$

As our basis and inductive steps hold, our proof is complete.

$\square$

The closed form from Theorem G.1 allows us to derive a conversion formula for $\alpha_2$ via our invariance formula in Equation 13.

$$
\begin{aligned}
\frac{1}{M_1} \sum_{i=1}^{M_1} t_e\left(\frac{N}{M_1}\right) &= \frac{1}{M_2} \sum_{i=1}^{M_2} t_e\left(\frac{N}{M_2}\right) \\
\frac{M_1}{M_1}\left(C - (C - min_f)(1 - \alpha_1)^{\frac{N}{M_1}}\right) &= \frac{M_2}{M_2}\left(C - (C - min_f)(1 - \alpha_2)^{\frac{N}{M_2}}\right) \\
(C - min_f)(1 - \alpha_1)^{\frac{N}{M_1}} &= (C - min_f)(1 - \alpha_2)^{\frac{N}{M_2}} \\
(1 - \alpha_1)^{\frac{N}{M_1}} &= (1 - \alpha_2)^{\frac{N}{M_2}} \\
(1 - \alpha_1)^{\frac{M_2}{M_1}} &= (1 - \alpha_2) \\
\alpha_2 &= 1 - (1 - \alpha_1)^{\frac{M_2}{M_1}}
\end{aligned}
\tag{18}
$$

We remark that our conversion formula is not dependent on the number of archive additions $N$.

Although our conversion formula assumes $f$ to be a constant objective, we conjecture that the formula holds generally for a convex objective $f$.

**Conjecture G.2.** *The archive learning rate conversion formula results in invariant behavior of CMA-MAE for two arbitrary archives $A_1$ and $A_2$ with archive resolutions $M_1$ and $M_2$, for a convex objective $f$.*

Our intuition is similar to the intuition behind Conjecture E.7, where we assume the elite hypervolume hypothesis holds (Vassiliades & Mouret, 2018). At the beginning of the CMA-MAE search, $f_A$ is a constant function and CMA-MAE optimizes for the global optimum, following the convergence properties of CMA-ES (Hansen & Ostermeier, 1997; Hansen et al., 2003). Eventually, the cells around the global optimum become saturated and the function $f - f_A$ forms a plateau around the global optimum. The invariance described in Eq. 13 implies that the $f_{A_1}$ and $f_{A_2}$ will increase at the same rate within the flat region of the plateau. Let $\theta_p$ be an arbitrary solution in the plateau and $\theta'$ be a solution on the frontier of the known hypervolume. The plateau of each archive $A_i$ expands when the solutions on the frontier of the elite hypervolume achieve a larger $f(\theta') - f_{A_i}(\theta')$ than the plateau $f(\theta_p) - f_{A_i}(\theta_p)$. We conjecture that the plateau will expand at the same rate in the two archives as $f_{A_1}$ and $f_{A_2}$ increase at the same rate for the plateau region, due to our invariance in Eq. 13.

We speculate that our conjecture explains why we observe invariant behavior across archive resolutions in the experiments of Section 7, even though $f$ is not a constant function in the linear projection and arm repertoire domains.

# H  A BATCH THRESHOLD UPDATE RULE FOR MAP-ANNEALING

In this paper, we presented an annealing method for updating a QD archive in the CMA-MAE algorithm, following the standard QD formulation where we add a single solution to the archive at a time. However, the recently developed QDax library (Lim et al., 2022) assumes that the updates to the archive happen in batch. In this section, we show that the archive update within a batch is dependent on the order that the solutions are processed. We then propose a candidate threshold update rule that is invariant to the order the solutions are processed within a batch update.

First, we show that the order solutions are added to the archive affects the current threshold update. Consider two solutions $\boldsymbol{\theta_1}$ and $\boldsymbol{\theta_2}$ that we add to the archive in a single batch. If $\boldsymbol{\theta_1}$ is added before $\boldsymbol{\theta_2}$, then the threshold update becomes $t'_e = (1-\alpha)[(1-\alpha)t_e + \alpha f(\boldsymbol{\theta_1})] + \alpha f(\boldsymbol{\theta_2}) = (1-\alpha)^2 t_e + (1-\alpha)\alpha f(\boldsymbol{\theta_1}) + \alpha f(\boldsymbol{\theta_2})$. If $\boldsymbol{\theta_2}$ is added before $\boldsymbol{\theta_1}$, then the threshold update becomes $t''_e = (1-\alpha)[(1-\alpha)t_e + \alpha f(\boldsymbol{\theta_2})] + \alpha f(\boldsymbol{\theta_1}) = (1-\alpha)^2 t_e + (1-\alpha)\alpha f(\boldsymbol{\theta_2}) + \alpha f(\boldsymbol{\theta_1})$.

To compare $t'_e$ to $t''_e$, we compute $t'_e - t''_e$:

$$
\begin{aligned}
t'_e - t''_e &= (1-\alpha)^2 t_e + (1-\alpha)\alpha f(\boldsymbol{\theta_1}) + \alpha f(\boldsymbol{\theta_2}) \\
&= [(1-\alpha)^2 t_e + (1-\alpha)\alpha f(\boldsymbol{\theta_2}) + \alpha f(\boldsymbol{\theta_1})] \\
&= (1-\alpha)\alpha[f(\boldsymbol{\theta_1}) - f(\boldsymbol{\theta_2})] + \alpha[f(\boldsymbol{\theta_2}) - f(\boldsymbol{\theta_1})] \\
&= [(1-\alpha)\alpha - \alpha][f(\boldsymbol{\theta_1}) - f(\boldsymbol{\theta_2})] \\
&= -\alpha^2[f(\boldsymbol{\theta_1}) - f(\boldsymbol{\theta_2})]
\end{aligned}
$$

From the above derivation, we see that the difference between thresholds is dependent on the solution's objective values when added to the archive in different order. This means when adding solutions to the archive in a batch, the update is dependent on the solution order in the batch.

Our goal is to make the threshold update invariant to the order the solutions are added to the archive. First, consider a subset of the batch that contains $c$ solutions all landing in the same cell of the archive and exceeding the current threshold $t_e$. Adding the solutions in batch order results in the following threshold update:

$$
t'_e = (1-\alpha)^c t_e + \sum_{j=1}^{c} (1-\alpha)^{c-j} \alpha f(\boldsymbol{\theta_j})
$$

Let $X$ be a random variable corresponding to the threshold for a given permutation of the batch. To become invariant to the batch addition, we will change the threshold update to be $\mathbb{E}[X]$, the expected value of $t'_e$ across all random permutations of the batch.

Let $X_i$ be a random variable corresponding to the contribution of only $f(\boldsymbol{\theta_i})$ to the threshold update and $Y$ be a constant random variable corresponding to the contribution of the previous threshold. As expectation is linear, we have $\mathbb{E}[X] = \mathbb{E}[Y] + \sum_{j=1}^{c} \mathbb{E}[X_j]$. Next, we compute the value of $\mathbb{E}[X_i]$ for an arbitrary solution $\boldsymbol{\theta_i}$ in the batch:

$$
\begin{aligned}
\mathbb{E}[X_i] &= \sum_{j=1}^{c} \Pr(\boldsymbol{\theta_i} \text{ is at position } j \text{ in the batch})(1-\alpha)^{c-j}\alpha f(\boldsymbol{\theta_i}) \\
&= \sum_{j=1}^{c} \frac{(c-1)!}{c!}(1-\alpha)^{c-j}\alpha f(\boldsymbol{\theta_i}) \\
&= \sum_{j=1}^{c} \frac{1}{c}(1-\alpha)^{c-j}\alpha f(\boldsymbol{\theta_i})
\end{aligned}
$$

Next, we rework $\mathbb{E}[X]$ into a simpler formula, where $f^* = \frac{\sum_{k=1}^{c} f(\boldsymbol{\theta_k})}{c}$:

$$\mathbb{E}[X] = \mathbb{E}[Y] + \sum_{j=1}^{c} \mathbb{E}[X_j]$$

$$= (1-\alpha)^c t_e + \sum_{k=1}^{c} \sum_{j=1}^{c} \frac{1}{c}(1-\alpha)^{c-j}\alpha f(\boldsymbol{\theta_k})$$

$$= (1-\alpha)^c t_e + \sum_{j=1}^{c} (1-\alpha)^{c-j}\alpha \frac{1}{c} \sum_{k=1}^{c} f(\boldsymbol{\theta_k})$$

$$= (1-\alpha)^c t_e + \sum_{j=1}^{c} (1-\alpha)^{c-j}\alpha f^*$$

$$= (1-\alpha)^c t_e + \alpha f^* \sum_{j=0}^{c-1} (1-\alpha)^j$$

$$= (1-\alpha)^c t_e + \alpha f^* \frac{1-(1-\alpha)^c}{\alpha}$$

$$= (1-\alpha)^c t_e + f^*(1-(1-\alpha)^c)$$

We propose the above expectation as the batch threshold update rule, where $f^*$ is the average objective value for all solutions in the batch that exceed the threshold $t_e$ for a given cell. We observe that the rule is independent of the solution order. Furthermore, if $\alpha = 0$, the update becomes $t_e$, and Theorem 5.1 still holds. If $\alpha = 1$, the update becomes $f^*$, which is the average of solutions that increase the threshold. We view this update as a smooth parallel addition compared to CMA-ME, which would add the best solution from the batch for any solution order. We leave exploring alternative batch update rules for future work.

# I ON IMPROVING THE QUALITY OF LATENT SPACE ILLUMINATION

We describe the limitations arising from the exact problem setup for our main experiments, adopted from previous work (Fontaine & Nikolaidis, 2021a), on producing high-quality images. We then discuss ideas from the generative art community for improving the setup and an additional experiment that incorporates these ideas to generate high-quality and diverse images.

## I.1 MAIN LSI EXPERIMENTS

In the main latent space illumination (LSI) experiments in section 6, we showed that CMA-MAE outperformed the other QD algorithms according to standard QD metrics following the exact setup of prior work (Fontaine & Nikolaidis, 2021a)

In the these experiments, we used latent space illumination as purely an optimization benchmark. However, obtaining high performance on LSI as a benchmark can be a competing objective with producing high quality images.

First, finding solutions that result in a high objective value does not always result in high quality images that match the text prompt. For example, a QD algorithm can find images that result in CLIP reporting a high similarity score by leaving the training distribution of StyleGAN.

Furthermore, we use the CLIP loss as a measure function, thus a QD algorithm attempts to both decrease and increase the loss function to cover the measure space. Increasing the loss function results in minimizing similarity with the text prompt, which can be attained by unrealistic images.

In the main LSI experiments all derivative-free QD algorithms would drift out of the latent distribution and produce archives of low image quality. We found that CMA-MAE would stay in the latent distribution longer before drifting out of distribution during exploration, due to the low archive learning rate $\alpha$ prioritizing the objective.

To address drifting out of distribution, we adopt a "timeout" restart rule proposed by other evolution strategy-based quality diversity algorithms (Colas et al., 2020; Paolo et al., 2021). A timeout restart rule runs for a fixed number of iterations before restarting. We add to the basic restart rule of CMA-MAE (Appendix A) an additional criterion for restarting based on the timeout restart rule. To generate all the LSI collages in section K, we use a timeout of 50 iterations for both CMA-MAE and CMA-MAEGA.

While we retained the same setup as in previous work for comparison purposes, we can change the setup by adopting ideas from the generative art community to produce very high quality images. We describe these ideas in section I.2.

## I.2  INNOVATIONS FROM THE GENERATIVE ART COMMUNITY

Beyond the specifics of the QD optimization algorithm, many aspects of latent space illumination can be improved. For example, prior work Frans et al. (2021) on guiding single-objective optimization with CLIP notes that the gradients that CLIP provides can be noisy and recommends data augmentations of the generated images, such as tiling or translating each image, before being passed to CLIP. This change can help smooth the gradients for gradient descent optimizers like Adam and can make the generated images retained by the archive match their text prompts more accurately.

Prior work on optimizing the latent space of VQ-GAN (Crowson et al., 2022) also notes that CLIP will not always provide smooth optimization gradients, nor accurate objective values. The authors recommend a different data augmentation, by creating a batch of random cutouts of the generated image and passing those images to CLIP, which produces smoother gradients and objective values. The paper also recommends regularizing the latent codes so that they become attracted to the Gaussian ball that captures the training disribution of the GAN. Both these techniques could improve the qualitative performance of latent space illumination.

Finally, we used the first version of StyleGAN (Karras et al., 2019) that was used in previous work (Fontaine & Nikolaidis, 2021a). Recent versions of StyleGAN (Karras et al., 2020b;a; 2021) can further improve the quality of the generated images.

We describe details of the improved setup in section I.3.

## I.3  IMPROVING QUALITY OF GENERATED IMAGES

To improve image quality, we include an additional experiment where we run each QD algorithm with a configuration inspired by the above findings from the generative art community.

First, we replace StyleGAN (Karras et al., 2019) with StyleGAN2 (Karras et al., 2020b), which produces better images and has a well-conditioned latent space for optimization.

Next, we change the latent space being optimized by QD. First note, that the StyleGAN architecture has multiple latent spaces to be optimized. StyleGAN consists of both a $z$-space latent space of size 512 and a mapping network that maps to 18 latent codes of size 512 at different levels of detail in the final image. This $18 \times 512$ tensor is known as $w$-space. The original LSI experiments of Fontaine & Nikolaidis (2021a) were based of a blogpost (Perez, 2021) that respresented the search space for LSI as a single 512 dimensional vector whose weights were shared for each level of detail in $w$-space. In this experiment, we will optimize the full $n = 18 \times 512 = 9216$ $w$-space with each QD algorithm for fine grain control of the generated images.

Instead of using restarts in the StyleGAN experiments to keep the search within latent space, we adopt the $w$-space regularization of Crowson et al. (2022). We compute an average $w$-space position by sampling $10^4$ points sampled from $\mathcal{N}(0, I)$ in $z$-space, then passing these points through the mapping network to find their position in $w$-space. We compute the standard error across each dimension. To regularize the latent space, we compute the distance from this $w$-space Gaussian distribution. If the distance from mean exceeds the Gaussian ball of highest density, we apply an L2 penalty to the objective $f$ to move the search back into the training distribution.

The LSI experiments from prior work (Fontaine & Nikolaidis, 2021a) downsample from the $1024 \times 1024$ images produced by StyleGAN to the $224 \times 224$ images required for input to the CLIP model. Following prior work (Crowson et al., 2022), we adopt the cutout technique that clips

32 images from the StyleGAN output of varying sizes, downsamples each cutout to $224 \times 224$, and passes each of these $224 \times 224$ cutouts to CLIP for evaluation. The loss becomes the average of the CLIP loss for all cutouts. This technique has been shown to smooth gradients for CLIP in single-objective latent space optimization.

Instead of starting the search at the latent code $\mathbf{0}$, we sample 512 latent codes from $\mathcal{N}(0, I)$ in $z$-space then select the image resulting in the highest objective value as the starting $w$-space latent code.

Finally, the prior LSI experiments leveraged the text prompt "A small child." as a proxy for age. However, this text prompt only specifies one end of the age measure. To correct this issue, we can pair a positive text prompt "A small child." with a negative text prompt "An elderly person." as a proxy for age. We compute the measure output by subtracting one CLIP loss from the other.

We run the improved LSI experiment with the text descriptor "A photo of the face of Tom Cruise." as an objective, the text pair "A photo of Tom Cruise as a small child." and "A photo of Tom Cruise as an elderly person." as a proxy measure for age, and "A photo of Tom Cruise with short hair." and "A photo of Tom Cruise with long hair." as a proxy measure for hair length.

Fig. 4 shows photos of Tom Cruise at varying hair lengths and ages, generated by the CMA-MAEGA algorithm in a single run.

## J   ON THE EFFECT OF THRESHOLD INITIALIZATION

In this paper we introduce two hyperparameters for our proposed CMA-MAE algorithm: the archive learning rate $\alpha$ and a threshold initialization $min_f$. In this section we discuss the effect of different $min_f$ initializations on the performance and behavior of CMA-MAE. Finally, we run an ablation on $min_f$, similar to the ablation on archive learning rate $\alpha$ in Section 7.

First, consider the effect of $min_f$ on the extreme cases of the CMA-MAE. When $\alpha = 0$, then according to Theorem E.1, CMA-MAE behaves identically to CMA-ES, and $min_f$ has no effect on the behavior of CMA-MAE. Conversely when $\alpha = 1$, then according to Theorem E.3, CMA-MAE behaves identically to CMA-ME when $min_f$ approaches an arbitrarily large negative number. As $min_f$ increases for $\alpha = 1$, CMA-MAE will rank some solutions that discover existing cells higher than solutions that discover new, empty cells, thus it will behave differently than CMA-ME.

Next, we discuss the behavior for $0 < \alpha < 1$. Recall the elite hypervolume hypothesis (Fefferman et al., 2016), which states that optimal solutions for the MAP-Elites archive form a connected region in search space, the elite hypervolume. According to the proof sketch of Conjecture E.7, early in the search CMA-MAE behaves identically to CMA-ES to find a solution point on the elite hypervolume. As the thresholds of cells around this solution point become saturated, the objective $f - f_A$ forms a plateau around the local optimum. Within the plateau, CMA-MAE triggers the density descent property of Theorem E.6 and evenly explores the known elite hypervolume until the plateau in $f - f_A$ dips below the frontier of the known hypervolume. This causes the known hypervolume to expand until all cells of the archive are filled.

Next, we discuss how the selection of $min_f$ affects the rate of expansion of the elite hypervolume. First, we consider two solution points: $\boldsymbol{\theta_1}$ represents a local optimum in the elite hypervolume and $\boldsymbol{\theta_2}$ represents a nearby point mapped to a different archive cell with a smaller objective value, or formally $f(\boldsymbol{\theta_1}) > f(\boldsymbol{\theta_2})$ and $\|\boldsymbol{\theta_1} - \boldsymbol{\theta_2}\|_2 \leq \epsilon$. We let $f_A(\boldsymbol{\theta_1}) = t_{e_1}$ and $f_A(\boldsymbol{\theta_2}) = t_{e_2}$, where $t_{e_1}$ and $t_{e_2}$ represent the thresholds of the cells that $\boldsymbol{\theta_1}$ and $\boldsymbol{\theta_2}$ map to, respectively. Both thresholds $t_{e_1}$ and $t_{e_2}$ are initialized with $min_f$.

We first examine the case where $min_f > f(\boldsymbol{\theta_1}) > f(\boldsymbol{\theta_2})$. Here, the thresholds $t_{e_1}$ and $t_{e_2}$ will not change and none of the two solutions points will be added to the archive. CMA-MAE will then only optimize for the objective value and behave identically to CMA-ES.

The second case is $f(\boldsymbol{\theta_1}) > min_f > f(\boldsymbol{\theta_2})$. Here, $\boldsymbol{\theta_2}$ will not get added to the archive and $t_{e_2} = min_f$ will not change, while $t_{e_1}$ will increase based on the update rule $t_{e_1} \leftarrow (1-\alpha)t_{e_1} + f(\boldsymbol{\theta_1})$. Recall that CMA-MAE ranks solutions based on improvement $\Delta_i = f(\boldsymbol{\theta_i}) - t_{e_i}$. We observe that $\Delta_2 = f(\boldsymbol{\theta_2}) - min_f < 0$, while $\Delta_1 = f(\boldsymbol{\theta_1}) - t_{e_1} > 0$, thus there is no incentive for CMA-MAE to optimize for $\boldsymbol{\theta_2}$ and it will instead optimize for the solution point $\boldsymbol{\theta_1}$ that has the highest objective

value. We observe that $min_f$ then acts as a constraint that prevent exploration of measure space regions with objective values below $min_f$.

Finally, we let $f(\boldsymbol{\theta_1}) > f(\boldsymbol{\theta_2}) > min_f$. Initially, $\boldsymbol{\theta_1}$ will be ranked higher than $\boldsymbol{\theta_2}$, since $\Delta_1 = f(\boldsymbol{\theta_1}) - t_{e_1} = f(\boldsymbol{\theta_1}) - min_f$ and $\Delta_2 = f(\boldsymbol{\theta_2}) - t_{e_2} = f(\boldsymbol{\theta_2}) - min_f$, thus $\Delta_1 > \Delta_2$. CMA-MAE will thus optimize for $\boldsymbol{\theta_1}$, but $\Delta_1$ will decrease because of the update rule $t_{e_1} \leftarrow (1 - \alpha)t_{e_1} + f(\boldsymbol{\theta_1})$.

Next, we compute how many steps it will take for $\Delta_1 = \Delta_2$. When $\Delta_1 = \Delta_2$, a plateau forms for $f - f_A$ and CMA-MAE transitions from optimizing like CMA-ES to expanding the frontier of the known hypervolume via density descent. We leverage Theorem G.1 that yields a closed form for updating a cell $k$ times for a fixed objective value $C$: $t_e(k) = C - (C - min_f)(1 - \alpha)^k$.

Let $k_1$ and $k_2$ be the number of times the cells containing $\boldsymbol{\theta_1}$ and $\boldsymbol{\theta_1}$ are sampled, respectively. We note that CMA-MAE behaves like CMA-ES until we reach the density descent property, therefore the cell containing $\boldsymbol{\theta_1}$ will be sampled more times than the cell containing $\boldsymbol{\theta_2}$ and $k_1 > k_2$, where the gap $k_1 - k_2$ grows as more optimization steps are taken.

$$\Delta_1 = \Delta_2$$
$$f(\boldsymbol{\theta_1}) - t_{e_1} = f(\boldsymbol{\theta_2}) - t_{e_2}$$
$$f(\boldsymbol{\theta_1}) - [f(\boldsymbol{\theta_1}) - (f(\boldsymbol{\theta_1}) - min_f)(1 - \alpha)^{k_1}] = f(\boldsymbol{\theta_2}) - [f(\boldsymbol{\theta_2}) - (f(\boldsymbol{\theta_2}) - min_f)(1 - \alpha)^{k_2}]$$
$$(f(\boldsymbol{\theta_1}) - min_f)(1 - \alpha)^{k_1} = (f(\boldsymbol{\theta_2}) - min_f)(1 - \alpha)^{k_2}$$
$$\frac{(1 - \alpha)^{k_1}}{(1 - \alpha)^{k_2}} = \frac{f(\boldsymbol{\theta_2}) - min_f}{f(\boldsymbol{\theta_1}) - min_f}$$
$$(1 - \alpha)^{k_1 - k_2} = \frac{f(\boldsymbol{\theta_2}) - min_f}{f(\boldsymbol{\theta_1}) - min_f}$$
$$k_1 - k_2 = \frac{\log \frac{f(\boldsymbol{\theta_2}) - min_f}{f(\boldsymbol{\theta_1}) - min_f}}{\log (1 - \alpha)}$$
$$k_1 - k_2 = \frac{\log \frac{f(\boldsymbol{\theta_1}) - min_f}{f(\boldsymbol{\theta_2}) - min_f}}{-\log (1 - \alpha)} \tag{19}$$

We note that $-\log (1 - \alpha)$ is a positive value as $1 - \alpha < 1$. We see that the number of times that $\boldsymbol{\theta_1}$ needs to be sampled more than $\boldsymbol{\theta_2}$ depends on the log ratio of the gaps between the objective values and $min_f$ and on the learning rate $\alpha$.

As $min_f$ decreases, number of optimization steps required to reach the plateau property approaches 0 asymptotically. While this shows that $min_f$ does have an effect on the behavior of the algorithm, since $min_f$ appears on the log ratio, we expect the effect of changing $min_f$ to be small.

We ran an ablations study by varying $min_f$ on the linear projection and arm repertoire domains. We explore different values of $min_f \in \{-80, -40, 0, 40, 80\}$. We ran each experimental setup for 20 trials each and report the results in Table 5.

We note that each domain remaps the objective values to the range $[0, 100]$. For $min_f$ smaller than the range, we observe that changing $min_f$ has a negligible effect on performance. On the other hand, positive values for $min_f$ constrain the search to solutions with $f \geq min_f$ (see Fig. 9), thus coverage decreases. These results match our theoretical analysis.

We note that in the LP (plateau) all optimal solutions for each cell are 100 and Arm Repertoire domain all optimal solutions for each cell are close to 100. Since all $min_f$ values in our range are below 100, we do not observe any effects on performance, even for positive values of $min_f$.

| $\min_f$ (CMA-MAE) | LP (sphere) | | LP (Rastrigin) | | LP (plateau) | | Arm Repertoire | |
|---|---|---|---|---|---|---|---|---|
| | QD-score | Coverage | QD-score | Coverage | QD-score | Coverage | QD-score | Coverage |
| -80 | 64.74 | 83.73% | 52.23 | 81.65% | 77.20 | 77.24% | 78.97 | 79.25% |
| -40 | 64.94 | 83.83% | 52.53 | 81.40% | 78.25 | 78.28% | 79.02 | 79.26% |
| 0 | 64.99 | 83.52% | 52.69 | 80.56% | 79.29 | 79.31% | 79.06 | 79.27% |
| 40 | 63.82 | 80.08% | 48.61 | 68.45% | 80.18 | 80.19% | 79.06 | 79.23% |
| 80 | 39.41 | 43.92% | 10.03 | 11.04% | 81.42 | 81.42% | 78.99 | 79.11% |

Table 5: Mean QD metrics after 10,000 iterations for CMA-MAE with varying $\min_f$ initialization.

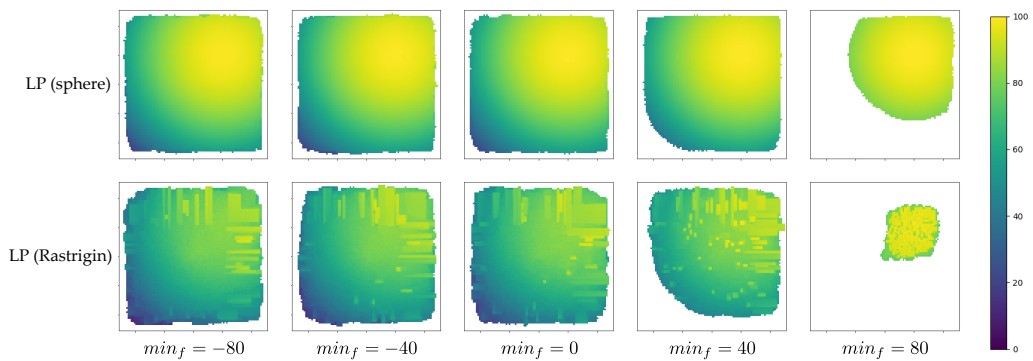

Figure 9: Example heatmaps from the $\min_f$ ablation. When $\min_f$ exceeds the objective value for solutions in the elite hypervolume, $\min_f$ acts as a constraint on exploration and CMA-MAE focuses on regions of the elite hypervolume that exceed $\min_f$ in objective value.

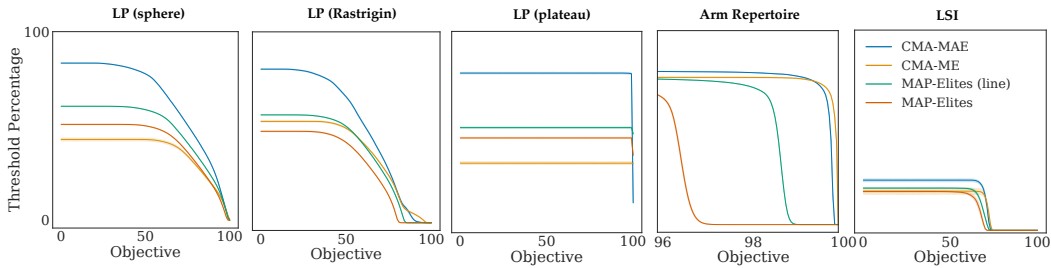

Figure 10: The percentage of cells in the y-axis with objective values larger than or equal to a threshold specified in the x-axis, with 95% confidence intervals. The percentage is the number of filled cells (filtered by the threshold) over the archive size. A larger area under each curve indicates better performance.

## K ADDITIONAL RESULTS

### K.1 GENERATED ARCHIVES AND ADDITIONAL METRICS

Table 6 presents the values of the QD-score, coverage, and best solution for each algorithm and domain. We used $\alpha = 0.01$ for CMA-MAE, identically to the main experiments. Similarly to Fontaine & Nikolaidis (2021a), we disambiguate the quality of solutions found and coverage by showing for MAP-Elites, MAP-Elites (line), CMA-ME and CMA-MAE the percentage of cells (y-axis) that have objective value greater than the threshold specified in the x-axis (Fig. 10).

### K.2 EXAMPLE ARCHIVES

Fig. 11 -15 show example archives for each algorithm and domain.

## Linear Projection (sphere)

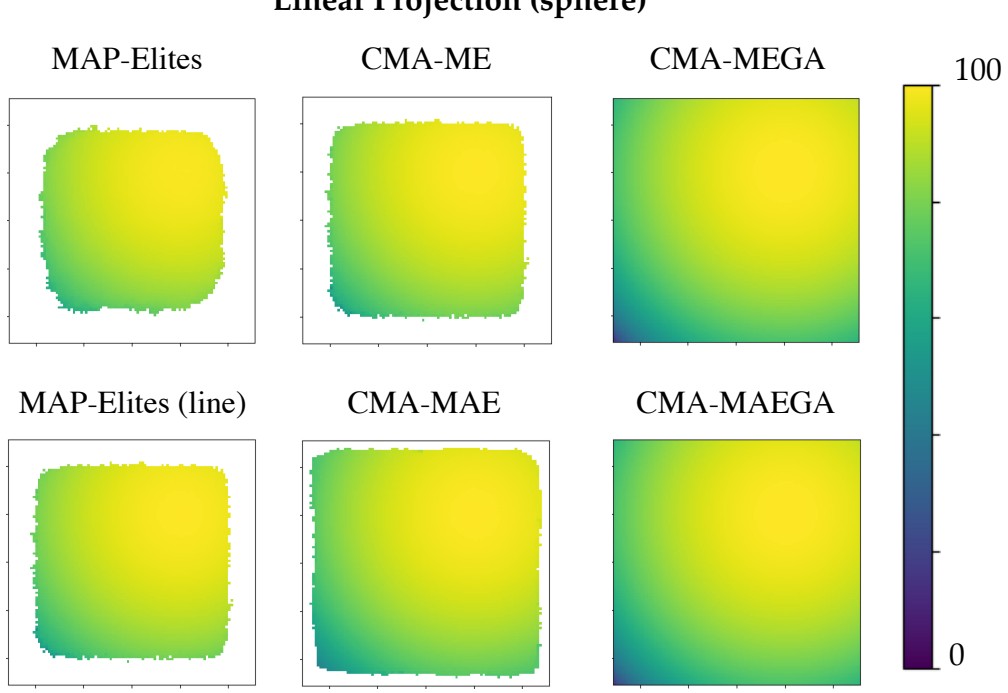

Figure 11: Example archives for each algorithm for the linear projection (sphere) domain.

## Linear Projection (Rastrigin)

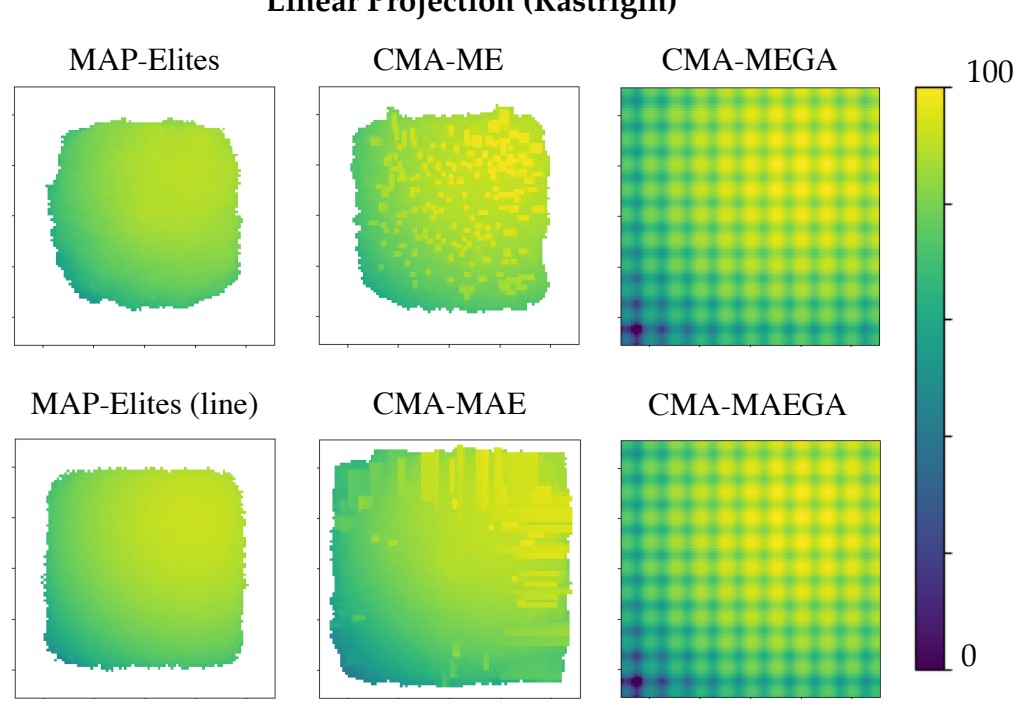

Figure 12: Example archives for each algorithm for the linear projection (Rastrigin) domain.

**Linear Projection (Plateau)**

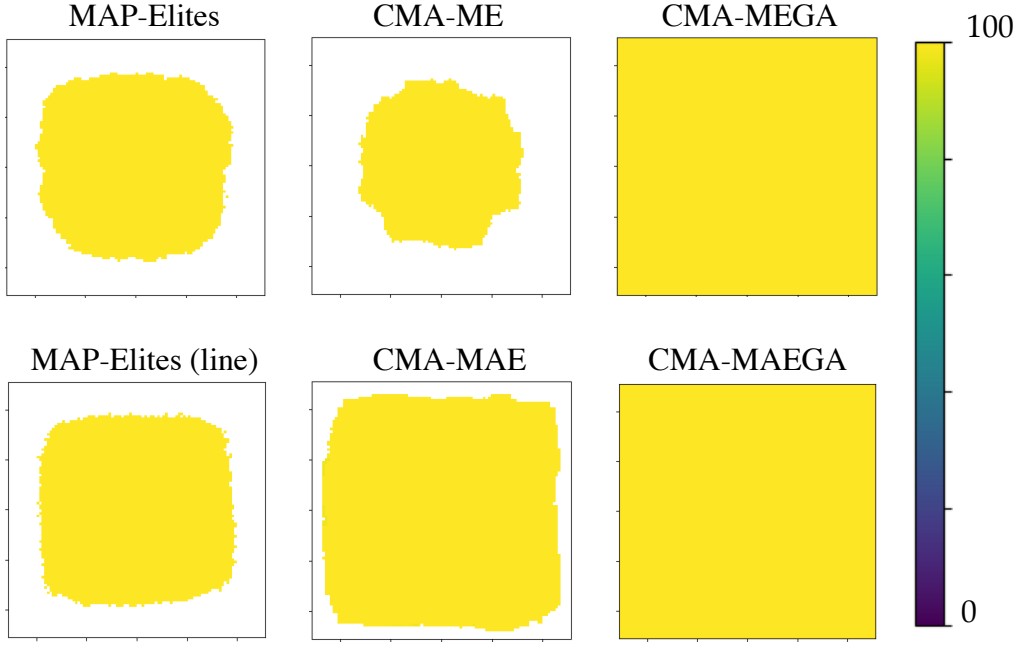

Figure 13: Example archives for each algorithm for the linear projection (plateau) domain.

**Arm Repertoire**

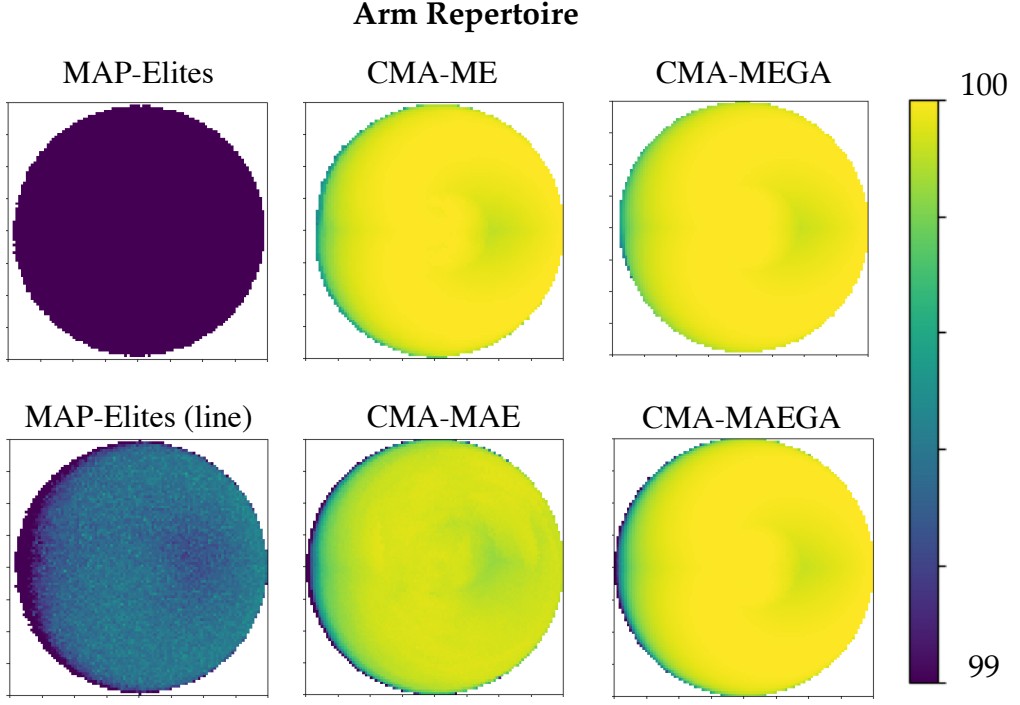

Figure 14: Example archives for each algorithm for the arm repertoire domain.

### K.3 Additional Experiments in the LSI Domain

We include the same additional experiments in the LSI (StyleGAN) domain as Fontaine & Nikolaidis (2021a). The first additional experiment has objective prompt "A photo of Jennifer Lopez" and

**Latent Space Illumination (StyleGAN)**

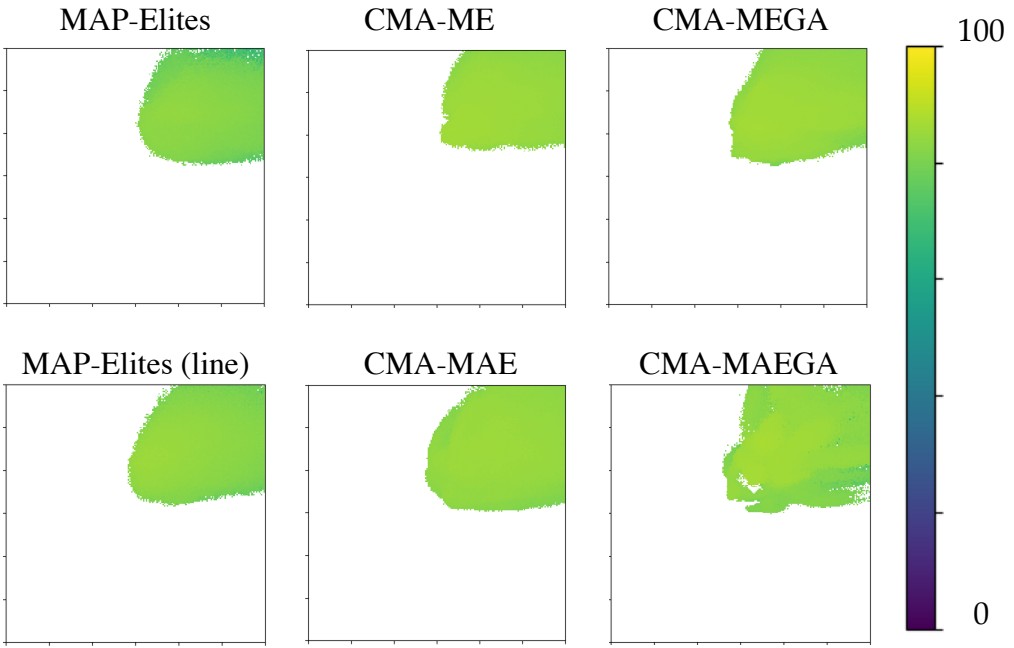

Figure 15: Example archives for each algorithm for the LSI (StyleGAN) domain.

**Latent Space Illumination (StyleGAN2)**

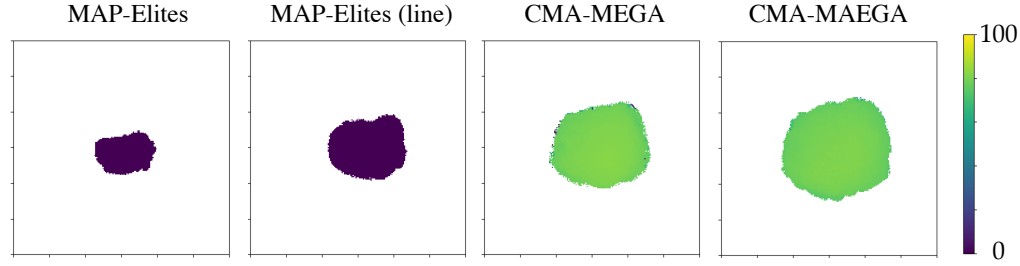

Figure 16: Example archives for each algorithm for the LSI (StyleGAN2) domain.

measure prompts "A small child." and "A woman with long blonde hair." The second has objective prompt "A photo of Elon Musk" and measure prompts "A person with red hair." and "A man with blue eyes." Table 7 shows the results of the additional runs, as well as the Beyoncé run, with objective prompt "A photo of Beyonce.", from the main paper.

### K.4    ADDITIONAL RESULTS FOR VARYING RESOLUTIONS

Fig. 17 shows the QD-score and coverage of CMA-MAE with resolution-dependent archive learning rate and the baselines, for each benchmark domain. For CMA-MAE, we set the resolution dependent archive learning rate $\alpha$ using the conversion formula from Appendix G, with $\alpha_1 = 0.01$ for resolution $100 \times 100$.

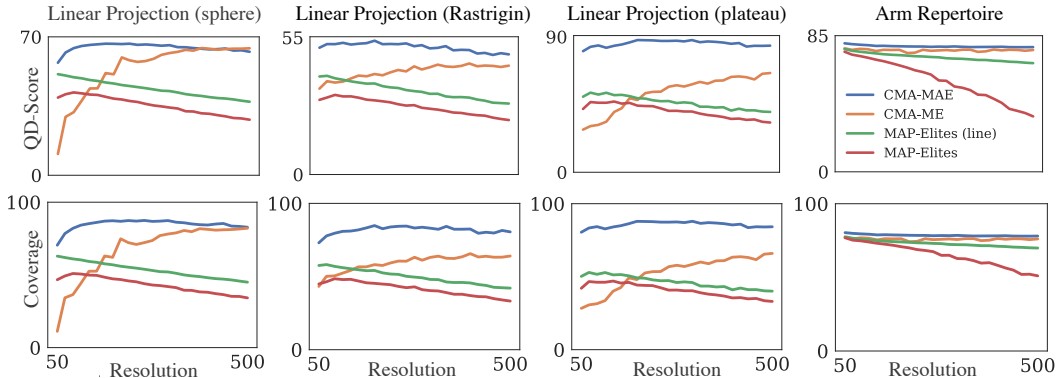

Figure 17: Final QD-score and coverage of each algorithm for 25 different archive resolutions.

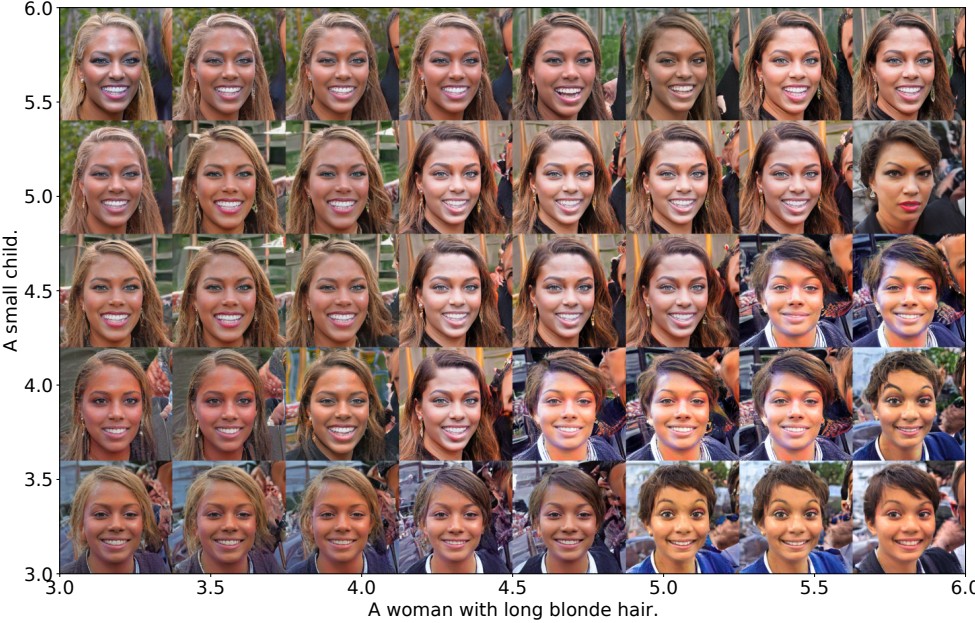

Figure 18: A latent space illumination collage generated by CMA-MAE for the objective "A photo of Beyonce." and for measures "A small child." and "A woman with long blonde hair." The axis values indicate the score returned by the CLIP model, where lower score indicates a better match.

## K.5 QUALITATIVE RESULTS IN THE LSI DOMAIN

We present example collages for CMA-MAE (Fig. 19, 20), 18) and for CMA-MAEGA (Fig. 21, 22) for the LSI (StyleGAN) domain. We also include collages of each run of all algorithms for all runs of LSI (StyleGAN) and LSI (StyleGAN2) in the anonymous Dropbox link: `https://www.dropbox.com/sh/7e22190k3p4zh69/AACcAKV7_Xgi4IMrhzxkCz5ca?dl=0`.

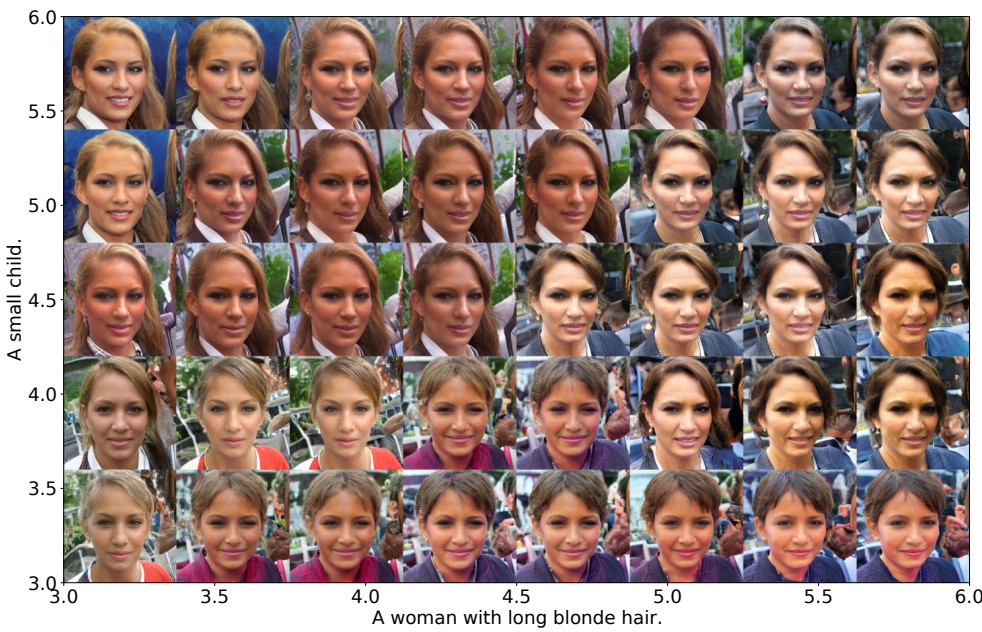

Figure 19: A latent space illumination (StyleGAN) collage generated by CMA-MAE for the objective "A photo of Jennifer Lopez." and for measures "A small child." and "A woman with long blonde hair." The axes values indicate the score returned by the CLIP model, where lower score indicates a better match.

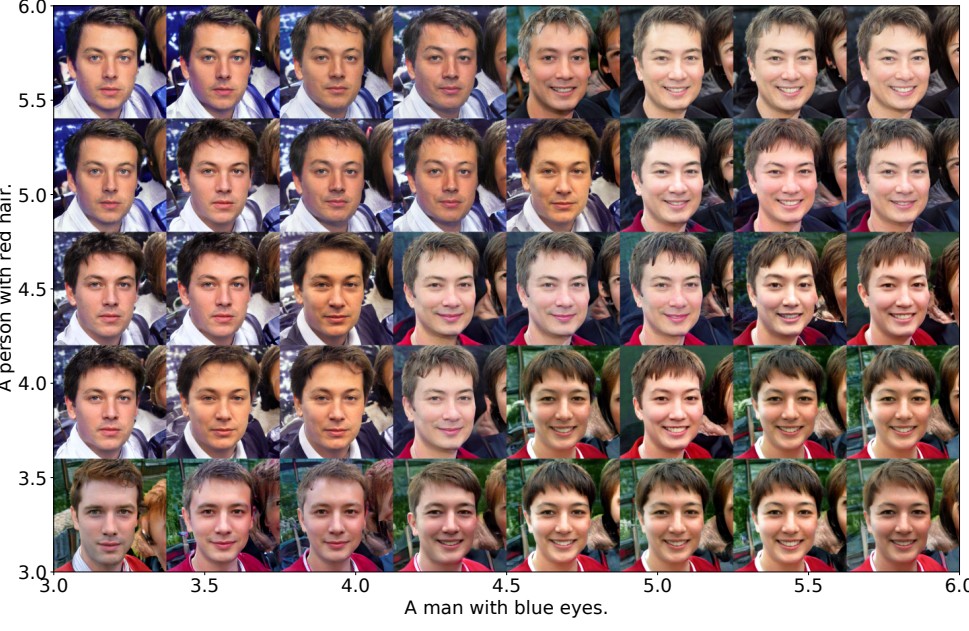

Figure 20: A latent space illumination (StyleGAN) collage generated by CMA-MAE for the objective "Elon Musk with short hair." and for measures "A man with blue eyes." and "A person with red hair." The axes indicate the score returned by the CLIP model, where lower score indicates a better match.

| | Linear Projection (sphere) | | |
|---|---|---|---|
| Algorithm | QD-score | Coverage | Best |
| MAP-Elites | $41.64 \pm 0.06$ | $50.80 \pm 0.09\%$ | $98.63 \pm 0.01$ |
| MAP-Elites (line) | $49.07 \pm 0.03$ | $60.42 \pm 0.05\%$ | $99.43 \pm 0.01$ |
| CMA-ME | $36.50 \pm 0.31$ | $42.82 \pm 0.40\%$ | $100.00 \pm 0.00$ |
| CMA-ME* | $45.07 \pm 0.07$ | $55.11 \pm 0.10\%$ | $99.23 \pm 0.02$ |
| CMA-ME (imp, opt) | $37.10 \pm 0.37$ | $43.62 \pm 0.45\%$ | $100.00 \pm 0.00$ |
| CMA-MAE | $64.86 \pm 0.04$ | $83.31 \pm 0.07\%$ | $99.59 \pm 0.01$ |
| CMA-MEGA | $75.32 \pm 0.00$ | $100.00 \pm 0.00\%$ | $100.00 \pm 0.00$ |
| CMA-MAEGA | $75.39 \pm 0.00$ | $100.00 \pm 0.00\%$ | $100.00 \pm 0.00$ |

| | Linear Projection (Rastrigin) | | |
|---|---|---|---|
| Algorithm | QD-score | Coverage | Best |
| MAP-Elites | $31.43 \pm 0.07$ | $47.88 \pm 0.12\%$ | $82.16 \pm 9,11$ |
| MAP-Elites (line) | $38.29 \pm 0.05$ | $56.51 \pm 0.09\%$ | $85.71 \pm 0.07$ |
| CMA-ME | $38.02 \pm 0.11$ | $53.09 \pm 0.16\%$ | $97.59 \pm 0.06$ |
| CMA-ME* | $35.06 \pm 0.05$ | $53.01 \pm 0.12\%$ | $83.47 \pm 0.08$ |
| CMA-ME (imp, opt) | $34.87 \pm 0.24$ | $48.93 \pm 0.40\%$ | $98.18 \pm 0.03$ |
| CMA-MAE | $52.65 \pm 0.06$ | $80.46 \pm 0.11\%$ | $95.90 \pm 0.25$ |
| CMA-MEGA | $63.07 \pm 0.00$ | $100.00 \pm 0.00\%$ | $100.00 \pm 0.00$ |
| CMA-MAEGA | $63.06 \pm 0.00$ | $100.00 \pm 0.00\%$ | $100.00 \pm 0.00$ |

| | Linear Projection (plateau) | | |
|---|---|---|---|
| Algorithm | QD-score | Coverage | Best |
| MAP-Elites | $47.07 \pm 0.17$ | $47.07 \pm 0.17\%$ | $100.00 \pm 0.00$ |
| MAP-Elites (line) | $52.20 \pm 0.19$ | $52.20 \pm 0.19\%$ | $100.00 \pm 0.00$ |
| CMA-ME | $34.54 \pm 0.35$ | $34.54 \pm 0.35\%$ | $100.00 \pm 0.00$ |
| CMA-ME* | $51.11 \pm 0.25$ | $51.11 \pm 0.25\%$ | $100.00 \pm 0.00$ |
| CMA-ME (imp, opt) | $31.91 \pm 0.43$ | $31.91 \pm 0.43\%$ | $100.00 \pm 0.00$ |
| CMA-MAE | $79.27 \pm 0.21$ | $79.29 \pm 0.21\%$ | $100.00 \pm 0.00$ |
| CMA-MEGA | $100.00 \pm 0.00$ | $100.00 \pm 0.00\%$ | $100.00 \pm 0.00$ |
| CMA-MAEGA | $100.00 \pm 0.00$ | $100.00 \pm 0.00\%$ | $100.00 \pm 0.00$ |

| | Arm Repertoire | | |
|---|---|---|---|
| Algorithm | QD-score | Coverage | Best |
| MAP-Elites | $71.40 \pm 0.03$ | $74.09 \pm 0.04\%$ | $97.38 \pm 0.03$ |
| MAP-Elites (line) | $74.55 \pm 0.02$ | $75.61 \pm 0.02\%$ | $99.16 \pm 0.01$ |
| CMA-ME | $75.82 \pm 0.11$ | $75.89 \pm 0.11\%$ | $100.00 \pm 0.00$ |
| CMA-ME* | $75.68 \pm 0.04$ | $76.13 \pm 0.03\%$ | $99.78 \pm 0.01$ |
| CMA-ME (imp, opt) | $75.91 \pm 0.07$ | $75.99 \pm 0.07\%$ | $100.00 \pm 0.00$ |
| CMA-MAE | $79.03 \pm 0.02$ | $79.24 \pm 0.02\%$ | $99.93 \pm 0.00$ |
| CMA-MEGA | $75.21 \pm 0.13$ | $75.25 \pm 0.13\%$ | $100.00 \pm 0.00$ |
| CMA-MAEGA | $79.27 \pm 0.02$ | $79.35 \pm 0.02\%$ | $100.00 \pm 0.00$ |

| | Latent Space Illumination (StyleGAN) | | |
|---|---|---|---|
| Algorithm | QD-score | Coverage | Best |
| MAP-Elites | $12.85 \pm 0.10$ | $19.42 \pm 0.16\%$ | $71.42 \pm 0.14$ |
| MAP-Elites (line) | $14.40 \pm 0.09$ | $21.11 \pm 0.11\%$ | $73.04 \pm 0.05$ |
| CMA-ME | $14.00 \pm 0.62$ | $19.57 \pm 0.90\%$ | $74.11 \pm 0.08$ |
| CMA-MAE | $17.67 \pm 0.27$ | $25.08 \pm 0.40\%$ | $73.48 \pm 0.18$ |
| CMA-MEGA | $16.08 \pm 0.37$ | $22.58 \pm 0.57\%$ | $74.95 \pm 0.27$ |
| CMA-MAEGA | $16.20 \pm 0.41$ | $23.83 \pm 0.46\%$ | $75.52 \pm 0.22$ |

| | Latent Space Illumination (StyleGAN2) | | |
|---|---|---|---|
| Algorithm | QD-score | Coverage | Best |
| MAP-Elites | $-276.18 \pm 32.00$ | $4.48 \pm 0.18\%$ | $-936.96 \pm 35.91$ |
| MAP-Elites (line) | $-827.25 \pm 25.99$ | $8.81 \pm 0.04\%$ | $-236.65 \pm 13.35$ |
| CMA-MEGA | $9.18 \pm 0.18$ | $14.91 \pm 0.12\%$ | $67.48 \pm 0.09$ |
| CMA-MAEGA | $11.51 \pm 0.09$ | $18.62 \pm 0.16\%$ | $66.17 \pm 0.08$ |

Table 6: Results: The QD-score, coverage, and best solution after 10,000 iterations for each algorithm and domain with standard errors. Larger values are better across all metrics.

| | LSI (StyleGAN): Beyoncé | | |
|---|---|---|---|
| Algorithm | QD-score | Coverage | Best |
| MAP-Elites | $12.85 \pm 0.10$ | $19.42 \pm 0.16\%$ | $71.42 \pm 0.14$ |
| MAP-Elites (line) | $14.40 \pm 0.09$ | $21.11 \pm 0.11\%$ | $73.04 \pm 0.05$ |
| CMA-ME | $14.00 \pm 0.62$ | $19.57 \pm 0.90\%$ | $74.11 \pm 0.08$ |
| CMA-MAE | $17.67 \pm 0.27$ | $25.08 \pm 0.40\%$ | $73.48 \pm 0.18$ |
| CMA-MEGA | $16.08 \pm 0.37$ | $22.58 \pm 0.57\%$ | $74.95 \pm 0.27$ |
| CMA-MAEGA | $16.20 \pm 0.41$ | $23.83 \pm 0.46\%$ | $75.52 \pm 0.22$ |

| | LSI (StyleGAN): Jennifer Lopez | | |
|---|---|---|---|
| Algorithm | QD-score | Coverage | Best |
| MAP-Elites | $12.51 \pm 0.28$ | $19.18 \pm 0.48\%$ | $70.87 \pm 0.27$ |
| MAP-Elites (line) | $14.73 \pm 0.06$ | $21.60 \pm 0.08\%$ | $73.50 \pm 0.13$ |
| CMA-ME | $15.24 \pm 0.37$ | $20.86 \pm 0.50\%$ | $75.39 \pm 0.09$ |
| CMA-MAE | $18.33 \pm 0.16$ | $25.42 \pm 0.24\%$ | $75.10 \pm 0.17$ |
| CMA-MEGA | $17.06 \pm 0.10$ | $23.40 \pm 0.14\%$ | $76.02 \pm 0.08$ |
| CMA-MAEGA | $16.45 \pm 0.27$ | $23.60 \pm 0.49\%$ | $76.42 \pm 0.13$ |

| | LSI (StyleGAN): Elon Musk | | |
|---|---|---|---|
| Algorithm | QD-score | Coverage | Best |
| MAP-Elites | $13.88 \pm 0.11$ | $23.15 \pm 0.14\%$ | $69.76 \pm 0.07$ |
| MAP-Elites (line) | $16.54 \pm 0.28$ | $25.73 \pm 0.31\%$ | $72.63 \pm 0.28$ |
| CMA-ME | $18.96 \pm 0.17$ | $26.18 \pm 0.24\%$ | $75.84 \pm 0.10$ |
| CMA-MAE | $22.10 \pm 0.31$ | $30.89 \pm 0.44\%$ | $75.25 \pm 0.20$ |
| CMA-MEGA | $21.82 \pm 0.18$ | $30.73 \pm 0.15\%$ | $76.89 \pm 0.15$ |
| CMA-MAEGA | $19.99 \pm 0.21$ | $30.12 \pm 0.42\%$ | $77.25 \pm 0.18$ |

Table 7: Results from additional runs for Beyoncé, Jennifer Lopez, and Elon Musk

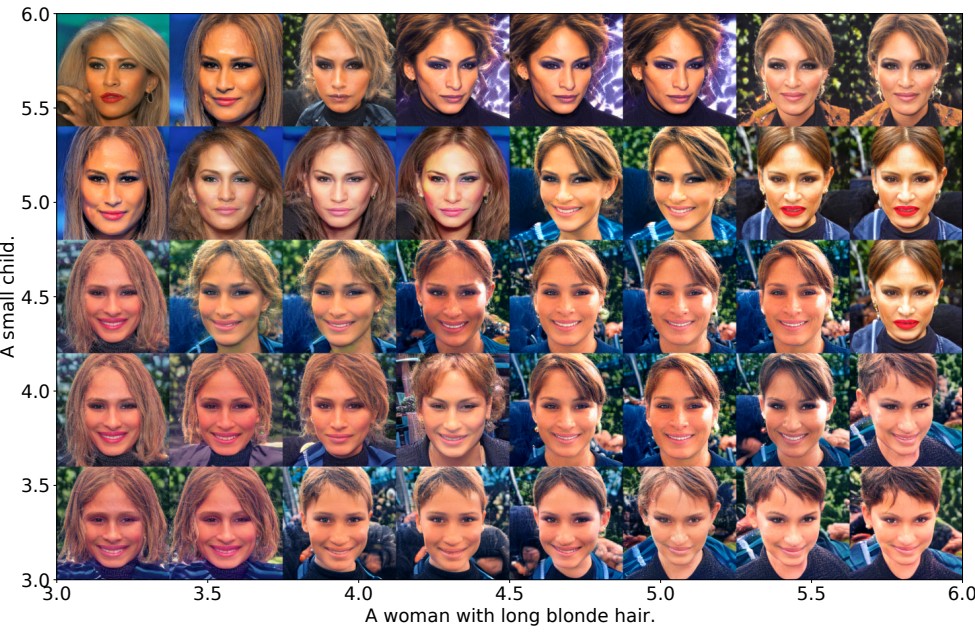

Figure 21: A latent space illumination (StyleGAN) collage generated by CMA-MAEGA for the objective "A photo of Jennifer Lopez." and for measures "A small child." and "A woman with long blonde hair." The axes values indicate the score returned by the CLIP model, where lower score indicates a better match.

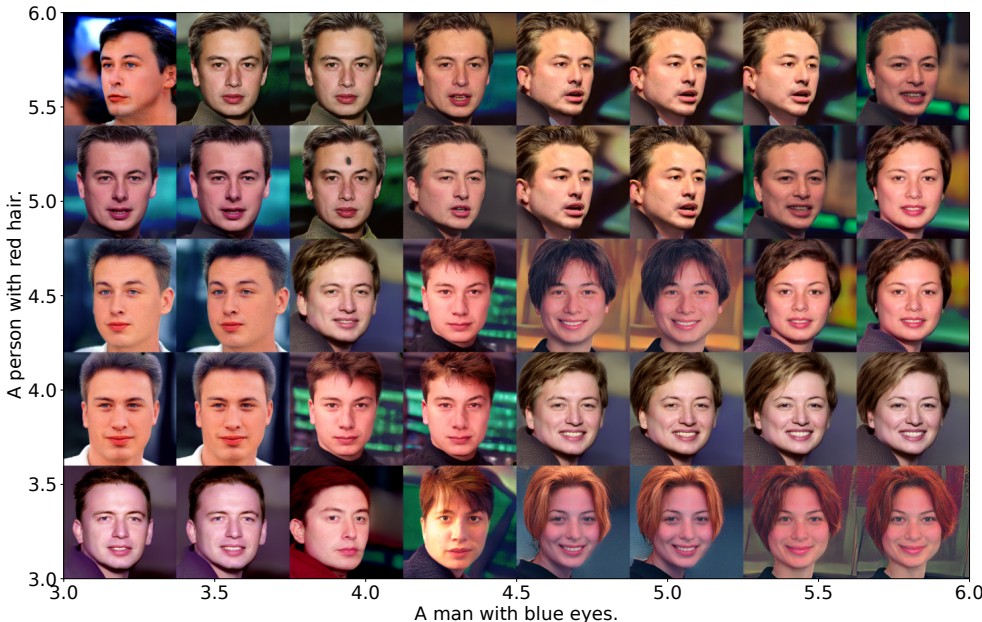

Figure 22: A latent space illumination (StyleGAN) collage generated by CMA-MAEGA for the objective "Elon Musk with short hair." and for measures "A man with blue eyes." and "A person with red hair." The axes values indicate the score returned by the CLIP model, where lower score indicates a better match.

