# OpenReview forum: "Covariance Matrix Adaptation MAP-Annealing"
_ICLR.cc/2023/Conference — Submitted to ICLR 2023_

### Official Review · Reviewer_prZ5 · 2022-10-23

**Confidence:** 5
**Correctness:** 3
**Technical Novelty And Significance:** 2
**Empirical Novelty And Significance:** 3
**Recommendation:** 3

**Clarity, Quality, Novelty And Reproducibility:**

The clarity, quality and reproducibility are good, but the novelty is very limited.

**Strength And Weaknesses:**

Strength

- The paper is overall well written and easy to understand.
- The experimental results show that the proposed method is robust and superior to other QD algorithms.

Weaknesses

- The proposed method is very incremental. Compared with CMA-ME, it just adds the smoothing factor to update the acceptance threshold.
- The experiments are not very convincing. The authors claimed that the modification allows CMA-MAE to spend more time on the promising zones before transitioning to exploration, thus can make CMA-MAE do better on domains with objectives that are hard to optimize as well as domains with flat objective functions. However, in the experiments, in addition to the common benchmarks, the authors only tested the methods on a simple function with the flat property, which makes the results not very convincing. I suggest the authors to conduct experiments on more problems with the mentioned properties.
- More ablation studies are needed. The performance of CMA-MAE depends on not only the setting of $\alpha$ but also that of $min_f$. What is the impact of the setting of $min_f$ on the performance?
- There are some minor problems, for example, in the section on limitations and future work, the order of the two paragraphs may be wrong.

**Summary Of The Paper:**

The paper introduces a new quality diversity algorithm, CMA-MAE, that try to address some limitations of the baseline algorithm CMA-ME. The authors introduce a learning rate $\alpha$ into the acceptance threshold, which encourages CMA-MAE to spend more time on the promising zones before transitioning to exploration. The performance of the proposed algorithm is evaluated in three QD benchmark domains and the results show that the proposed algorithm outperforms state-of-the-art QD algorithms.

**Summary Of The Review:**

The paper is overall well written and easy to understand. However, the proposed method is very incremental compared with the baseline algorithm, and more convincing experiments are needed to demonstrate the advantages of the method.

---

> ### Author Response · Authors · 2022-11-13
> **Response to Reviewer prZ5**
>
> > The experiments are not very convincing. The authors claimed that the modification allows CMA-MAE to spend more time on the promising zones before transitioning to exploration, thus can make CMA-MAE do better on domains with objectives that are hard to optimize as well as domains with flat objective functions. However, in the experiments, in addition to the common benchmarks, the authors only tested the methods on a simple function with the flat property, which makes the results not very convincing. I suggest the authors to conduct experiments on more problems with the mentioned properties.
>
> In our paper, we ran 6 total domains (sphere, rastrigin, plateau, arm, LSI (StyleGAN), LSI (StyleGAN2), against 8 total algorithms.
>
> 4 of these domains were presented in the first DQD paper (see “Differentiable Quality Diversity”), and we selected these domains for comparison purposes. In our paper we wish to test both QD and DQD algorithms, where the proposed DQD algorithm CMA-MAEGA and the relevant experiments are described in the appendix. For consistency and comparability, we wish to have the same domains for both QD and DQD algorithms. The original DQD paper has the linear projection, arm repertoire, and LSI domains; these are the only existing DQD benchmark domains to the best of our knowledge, and we use the same domains here.
>
> To evaluate the performance benefit on flat objectives, we introduced the plateau domain which causes CMA-ME to restart each iteration due to the natural gradient being 0. This validates empirically that CMA-MAE can explore on plateaus, and empirically supports the theory presented in Theorem 5.4.
>
> Regarding hard to optimize objectives, we state that a limitation of CMA-ME is that in these domains, CMA-ME abandons the objective too soon which harms performance. Our theory shows that CMA-MAE addresses this limitation by smoothly blending between CMA-ES, which solely optimizes the objective, and CMA-ME, which prioritizes exploration. We also refer reviewer prZ5 to the supplemental material where we include videos of CMA-MAE spending more time in high objective regions before transitioning to other regions of measure space. We note that the LSI (StyleGAN2) domain described in the appendix is very high-dimensional (n=9216) and hard to optimize, and CMA-MAEGA performs significantly better than CMA-MEGA in this domain.
>
> Overall, the paper contains 20 total experimental settings, including our additional robustness and archive resolution experiments, and a total of 1950 trial runs across all experiments. This is far more thorough than prior work on the DQD subject (see “Differentiable Quality Diversity”) and the CMA-ME paper. We believe that more experiments fall beyond the scope of a conference paper.
>
> > More ablation studies are needed. The performance of CMA-MAE depends on not only the setting of $\alpha$ but also that of $min_f$. What is the impact of the setting of $min_f$ on the performance?
>
> We added Appendix J to the revised paper to address this concern and thank reviewer prZ5 for the suggestion.
>
> > There are some minor problems, for example, in the section on limitations and future work, the order of the two paragraphs may be wrong.
>
> Can reviewer prZ5 elaborate? The paragraphs appear to be in the correct order. We first describe the limitations of CMA-MAE and then describe future work.

---

> > ### Comment · Reviewer_prZ5 · 2022-12-06
> > **Thanks for your reply**
> >
> > Thanks for your clarification on the experiments. But considering the novelty (which just introduces a parameter into an existing method), I keep the score as is. The theory is also very straightforward, as another reviewer also indicated.

---

### Official Review · Reviewer_KKju · 2022-10-25

**Confidence:** 3
**Correctness:** 3
**Technical Novelty And Significance:** 2
**Empirical Novelty And Significance:** 2
**Recommendation:** 5

**Clarity, Quality, Novelty And Reproducibility:**

The paper is easy to follow. Because of the proposal is incremental and the change is minimal, its novelty is limited. The experimental details are provided.

**Details Of Ethics Concerns:**

Section 2 and 3 are very similar to an existing paper [*]. It is not a copy-paste as each sentences are rephrased. Therefore, it may be fine. However, the structure of these sections are very close.

[*] Matthew Fontaine and Stefanos Nikolaidis. Differentiable quality diversity. NeurIPS 2021.

**Strength And Weaknesses:**

# Strength

* A minimal change to the existing approach, CMA-ME, has been proposed, but a statistically significant improvement has been observed in the benchmark problems.

# Weaknesses

Technical contribution: A simple approach is nice. However, at the same time, the proposal is incremental and its difference is minimal.

Test problem choice: It has not been discussed why these 5 test problems are selected. Because the selection of test problems is a very important part of the experimental design, this must be carefully explained. In this work, why are these 5 test problems are sufficient to reveal the goodness and limitation of the proposed approaches?

Discussion on the significance: I understand that the results are statistically significant. However, because I am not very familiar with QD community, it is hard to see whether this improvement is meaningful. A discussion on how meaningful these improvements are would improve the readers' understanding, especially for those who are not familiar with QD, which I guess the most audiences in this conference are.

Significance of this topic: Because the test problems are selected from the existing work where differentiability is assumed, a possible application of the proposed approach is not clear.


**Summary Of The Paper:**

This paper aims at improving CMA-ME, a quality-diversity (QD) algorithm. The authors focus on the way that the CMA-ME ranks candidate solutions. The authors hypothesize that the rapid change of the fitness threshold value, denoted by t_e, is the cause of three limitation highlighted in the QD community: prematurely abandoning the objective, struggling to explore flat objectives, and having poor performance for low-resolution archives. The proposed approach changes only the way to update the threshold: introducing a damping factor.

The proposed approach, CMA-MAE, is compared with CMA-ME, MAP-Elites (line) and MAP-Elites on 5 test problems. Higher QD-score and coverage are observed for all test problems.



**Summary Of The Review:**

It is hard to judge this paper. The technical contribution is minimal. However, this minimal change leads to a statistically significant improvement over the existing approaches. It is not clear how it contributes to ML communities. Probably because of the lack of my knowledge in QD, the significance of the proposed approach may be underestimated. I would currently suggest weak reject, but it may be increased if the authors response convinces me.

---

> ### Author Response · Authors · 2022-11-13
> **Response to Reviewer KKju**
>
> > Test problem choice: It has not been discussed why these 5 test problems are selected [...] In this work, why are these 5 test problems are sufficient to reveal the goodness and limitation of the proposed approaches?
>
> In our paper, we wish to test both QD and DQD algorithms, where the proposed DQD algorithm CMA-MAEGA and the relevant experiments are described in the appendix. For consistency and comparability, we wish to have the same domains for both QD and DQD algorithms. The original DQD paper has the linear projection, arm repertoire and LSI domains; these are the only existing DQD benchmark domains to the best of our knowledge and we use the same domains here. We note that in addition to these domains, we also added the plateau domain, which did not appear in the first DQD paper “Differentiable Quality Diversity”, to show how CMA-MAE addresses the flat objective limitation. We additionally added an LSI domain with StyleGAN2, which is a much harder domain to optimize. We show that our approach results in significant improvements in the QD score and coverage metrics in this new domain. We discuss our domain choices in detail in Appendix B.
>
> > Discussion on the significance: I understand that the results are statistically significant [...] A discussion on how meaningful these improvements are would improve the readers' understanding.
>
> In this paper, our proposed algorithms improve on the standard QD-score and coverage metrics on all domains, over the baseline algorithms. QD-score represents how well the QD algorithm filled the archive with high quality solutions, while the coverage metric reports how well the QD algorithm explored measure space. However, reducing a QD algorithm’s performance to a single scalar does not give a full picture of performance between algorithms. In figure 9 in the appendix, we report CDF plots that report how many solutions in each archive achieve an objective above the threshold defined by the $x$-axis. In other words, for a given objective value we can know how many solutions in the archive were at least as good as that objective value.
>
> From the results, the paper shows that CMA-MAE is the state-of-the-art derivative-free QD algorithm, and CMA-MAEGA is the state-of-the-art DQD algorithm.
>
> However, we believe the larger impact will be the robustness guarantees of the algorithm and ease of tuning. Current QD algorithms require carefully tuning the archive resolution hyperparameter as seen by figure 5 in the paper. However, CMA-MAE’s performance is invariant to archive resolution. The density descent property of Theorem 5.4 ensures that CMA-MAE will continue to explore on flat objectives, instead of restarting like CMA-ME. We believe that the robustness of CMA-MAE will make it easier to apply QD optimization to new domains that are too expensive to run extensive tuning experiments on. We anticipate the effect to be similar to how WGAN greatly improved the stability and robustness of GAN training, allowing GANs to be applied to a wider variety of datasets.
>
> > Significance of this topic: Because the test problems are selected from the existing work where differentiability is assumed, a possible application of the proposed approach is not clear.
>
> We chose domains with differentiable $f$ and $m$ to evaluate our proposed CMA-MAEGA algorithm. However, there was no space to present both CMA-MAE and CMA-MAEGA in the main paper. So we chose to just present CMA-MAE in the main paper and we moved the details and experiments involving CMA-MAEGA to the appendix.
>
> Both algorithms have multiple applications. CMA-ME has been applied to a large collection of problems including reinforcement learning, environment generation, and procedural content generation. To our knowledge, this is only the second paper with DQD domains, so the impact of DQD is currently limited to the domains in the paper. However, we believe that DQD algorithms will eventually have the breadth of scope of first-order optimization methods like gradient descent. Given gradient descent’s large impact on ML, we believe there is potential for a large impact on ML by advancing DQD algorithms.
>
> > Section 2 and 3 are very similar to an existing paper [...] the structure of these sections are very close.
>
> We solve the same problem definitions for QD and DQD presented in *. At the start of section 2, we mention where the problem definitions came from. Because we are solving the same problem, our experimental settings mirror * and share the same baselines, so that we can compare with the state-of-the-art in the presented domains.
>
> However, that is where the similarities between the two papers end. Our paper focuses on the theoretical properties of our new approach, which are all novel. We also focus on the robustness of our algorithm and invariance to archive resolutions, which were not experimental settings in *. Finally, we expand upon the experimental domains of * by adding the plateau and LSI (StyleGAN2) domains to our paper.

---

> > ### Comment · Reviewer_KKju · 2022-12-06
> > **Thanks for the response**
> >
> > I appreciate your response. The comments on the test problem choice and the discussion on the significance doesn't really answer my questions. For the significance of this topic, the answer is not satisfactory. Why don't you just select tasks that QD approaches (in particular CMA-ME variants) are more promising compared to other approaches for evaluation? If RL is a good application, why don't you apply it on RL and compare the performance with some other approaches? I suggest the authors to reconsider the organization of the experiments to show not only the goodness of the proposed approach over CMA-ME but also the goodness of the proposed (QD) approach over other approaches in a specific domain.

---

### Official Review · Reviewer_9oqX · 2022-10-30

**Confidence:** 3
**Correctness:** 3
**Technical Novelty And Significance:** 2
**Empirical Novelty And Significance:** 2
**Recommendation:** 5

**Clarity, Quality, Novelty And Reproducibility:**

Given the earlier works on CMA-ME and QD problems, the novelty seems less than expected for ICLR. The linguistic quality of the paper needs very significant improvement.

The reproducibility aspect is appreciable, the authors released detailed source codes.

**Strength And Weaknesses:**

Strength:

1) The paper presents a systematic study with adequate empirical experiments.

Weakness:

1) Significance and relevance of QD problem in machine learning along with some warm-up examples should have been given in the introduction.

2) The simple algorithmic change in CMA-ME is seemingly not significant enough to warrant a publication in ICLR. The theoretical proofs seem quite standard and straightforward.

3) For Figure 3, why are you plotting the QD score of different algorithms against number of iterations? WOn't it be such that these different algorithms will perform different amounts of jobs in their inner loops and hence, their "iterations" will not consume same amount of time?

4) The proposal was not adequatey validated through comparison against evolutionary algorithms of different genre than CMA-ES, like DE for QD problems. Also the multi-objective optimization approaches like the following one should have been compared:

Thomas Pierrot, Guillaume Richard, Karim Beguir, and Antoine Cully. 2022. Multi-objective quality diversity optimization. In Proceedings of the Genetic and Evolutionary Computation Conference (GECCO '22). Association for Computing Machinery, New York, NY, USA, 139–147. https://doi.org/10.1145/3512290.3528823

5) Why the parametric ANOVA test was used instead of non-parametric statistical tests?

6) Despite focusing on derivative-free QD algorithms, did you really handle a non-dfferentiable function like Weirstrass? Or I am missing something here?



**Summary Of The Paper:**

The paper proposes a variant of an already existing algorithm CMA-ME for Qualty-Diversity (QD) problems by introducing a learning rate based annealing function.

**Summary Of The Review:**

The paper presents an interesting study with appreciable set of experiments. However, the novelty and innovative content seem somewhat less for a conference like ICLR.

---

> ### Author Response · Authors · 2022-11-13
> **Response to Reviewer 9oqX**
>
> > Significance and relevance of QD problem in machine learning along with some warm-up examples should have been given in the introduction.
>
> We note the introduction contains a running machine learning example of searching a StyleGAN2+CLIP pipeline with a QD algorithm to generate a diverse collection of images that match a target prompt, but vary according to age and hair length. We note that targeted data synthesis from an ML model is an important machine learning problem, and quality diversity optimization is one way to solve this problem. We provide additional ML problems where quality diversity optimization has been applied in section 9, paragraph 2.
>
> > For Figure 3, why are you plotting the QD score of different algorithms against number of iterations? WOn't it be such that these different algorithms will perform different amounts of jobs in their inner loops and hence, their "iterations" will not consume same amount of time?
>
> We note that in each iteration, each algorithm processes the same batch size of solutions. Iterations in this paper are equivalent to epochs in training deep neural networks, which also operate over batches.
>
> > The proposal was not adequatey validated through comparison against evolutionary algorithms of different genre than CMA-ES, like DE for QD problems. Also the multi-objective optimization approaches like the following one should have been compared:
>
> The paper cited does not solve the same problem defined in section 2 and instead assumes more than one objective function $f_i$. Running MOME on our domains would give the same results as standard MAP-Elites on the experimental domains, since there is only one objective. Exploring multi-objective variants of CMA-MAE is an exciting future research direction, but beyond the scope of this work.
>
> Many QD algorithms exist, such as NSLC, SAIL, SERENE, and SHINE. However, we chose a large, but inexhaustive, subset of baselines most similar to our proposed approach. We did not include DE variants of MAP-Elites, because the algorithm is not currently supported in QD optimization libraries like pyribs or QDax and has complex tuning and implementation details.
>
> > Despite focusing on derivative-free QD algorithms, did you really handle a non-dfferentiable function like Weirstrass? Or I am missing something here?
>
> In the main paper we explore the derivative-free QD problem. This problem assumes that the QD algorithm does not have access to the gradients of f and m. In the appendix, we evaluate DQD problems which assume the QD algorithm has access to the gradients of $f$ and $m$, as we can augment the CMA-MEGA algorithm with our annealing approach to derive CMA-MAEGA. As a result, we chose domains with differentiable objectives and measures to provide consistent evaluation of both first-order and derivative-free methods.
>
> > The linguistic quality of the paper needs very significant improvement.
>
> Can reviewer 9oqX be more specific about the places for improvement? Reviewer prZ5 said the paper was “overall well written and easy to understand”. We want to improve the writing of the paper, but we need specifics from reviewer 9oqX on what is lacking.

---

> > ### Comment · Reviewer_9oqX · 2022-12-06
> > **Response to your rebuttal**
> >
> > While your response settles some of my concerns, the major problem of this paper is its incremental technical contribution that is also not innovative enough. Hence, I will keep my score unaltered.

---

### Author Response · Authors · 2022-11-13
**Response to all: Thank you very much for the reviews.**

We would like to thank the reviewers for taking the time to review our paper and provide feedback.

We would like to discuss why we are confident that this paper will be impactful, and why advancing the state-of-the-art in quality diversity (QD) optimization in both performance and robustness will bring value not just to the QD community, but also to the broader machine learning community.

While single-objective (SO) optimization has long been studied by the ICLR community to train deep learning models, optimization to generate datasets has been significantly less studied. At the same time, there is an increasing importance on data-centric AI and a focus on efficient synthesis of diverse and high-quality data for model training and testing.

QD optimization is particularly suited for dataset generation, because the algorithms produce a diverse collection of solutions, rather than SO optimization producing a single locally optimal solution. However, QD optimization has yet to have SO optimization level impact in ML, because the current algorithms are not robust enough with respect to their hyperparameters: MAP-Elites is non-adaptive and poorly chosen sigma or resolution can cause the algorithm to fail. CMA-ME is adaptive, but struggles to optimize low resolution archives and fails on flat objective regions.

CMA-MAE is the first QD algorithm to be invariant to archive resolution, it is an adaptive algorithm, and performs significantly better than all current QD baselines, setting a new state-of-the-art in QD optimization. We view this as a major step towards QD optimization being an effective tool for dataset synthesis.

Unfortunately, because the algorithmic change between CMA-ME and CMA-MAE is simple, the reviews reflect the perception that there is a small difference in the algorithmic behavior between the two algorithms and therefore the contribution of the paper is incremental.

We argue that this is not the case. CMA-MAE is derived from a simple modification of CMA-ME, but the effects of that change in the algorithm dynamics are complex and subtle. Explaining these dynamics requires a set of theoretical steps that could not fit into the main paper. We would encourage the reviewers and readers of the manuscript to carefully review our theory, which explains the complex dynamics of the new algorithm. In particular, the conjecture E.7 and proof sketch discusses why the improved exploration of CMA-MAE generalizes to multiple domains, while Appendix G proves CMA-MAE’s invariance to archive resolution.

We note that we could have obtained the same performance benefits with a much more complex algorithm. However, we found that we could satisfy all properties with a much simpler change, and we prefer to present a simpler algorithm, even though we were aware that a simple change may be perceived as incremental. We do consider this to be a strength, rather than a weakness of our approach.

Another criticism is that the selected domains do not show the strength of CMA-MAE. To show robustness we needed to run experiments in 20 experimental settings with a total of 1950 trials. We selected four QD benchmarks domains from the community that can be run efficiently to demonstrate robustness and performance of our derivative-free CMA-MAE algorithm. We introduced a new domain with a flat objective, and a much harder variant of the
latent space illumination domain presented in previous work, where we replaced StyleGAN with StyleGAN2 that has 9216 latent space parameters. Appendix D shows that the DQD variant of CMA-MAE, CMA-MAEGA significantly outperforms the CMA-ME variant CMA-MEGA on this domain.

We updated the paper with Appendix J to include the ablation requested by Reviewer prZ5. We respond to each reviewer and look forward to more discussion on the paper.

---

### Decision · Program_Chairs · 2023-01-20

**Decision:**

Reject

**Justification For Why Not Higher Score:**

See the above-mentioned major concern (A).

**Justification For Why Not Lower Score:**

N/A.

**Metareview: Summary, Strengths And Weaknesses:**

The main contribution of this work lies in proposing a variant of the known CMA-ME algorithm for quality diversity problems by introducing a learning rate into the acceptance threshold, so as to overcome the limitations of CMA-ME.

(A) While the empirical results presented in this paper show the better performance of their proposed modification, the reviewers all agree that the contribution (in particular, the algorithmic innovation and proofs of theoretical results) is incremental, even after reading the authors' rebuttal.

Besides the reviewers' suggestions for improvement, the authors can consider the following additional suggestions:

(1) If the authors think that the challenge lies in proving the theoretical results, they are advised to reorganize their paper to focus on explicitly explaining and highlighting the nontrivial challenges with proving the results. For example, are there important intermediate results that are not straightforward to be proven?

(2) The authors can consider whether the more complex algorithm with the same performance benefits leads to further insights not gleaned through the simple modification in this paper. If so, the authors can prove their equivalence and show the additional insights entailed by the more complex algorithm.